# REWARD IS ENOUGH: LLMS ARE IN-CONTEXT REINFORCEMENT LEARNERS

**Kefan Song**[*]
University of Virginia

**Amir Moeini**[*]
University of Virginia

**Peng Wang**
University of Virginia

**Lei Gong**
University of Virginia

**Rohan Chandra**
University of Virginia

**Shangtong Zhang**[†]
University of Virginia

**Yanjun Qi**[†]
University of Virginia

## ABSTRACT

Reinforcement learning (RL) is a framework for solving sequential decision-making problems. In this work, we demonstrate that, surprisingly, RL emerges during the inference time of large language models (LLMs), a phenomenon we term in-context RL (ICRL). To reveal this capability, we introduce a simple multi-round prompting framework, we call ICRL prompting, for inference-time self-improvement. The goal of ICRL prompting is to guide LLMs to perform reinforcement learning during inference for self-improvement on a given task. After each response, the model receives numerical scalar feedback, denoted as a reward. In the next round, we prompt the LLM again together with a context that concatenates all prior responses and their associated rewards. We consistently observe that response quality improves as the context grows. In other words, the LLM can optimize scalar reward signals during inference, exhibiting behavior analogous to reinforcement learning. We evaluate ICRL prompting on Game of 24, creative writing, ScienceWorld, and Olympiad-level math competitions (AIME and HMMT), demonstrating significant improvements over baselines such as Self-Refine and Reflexion. Notably, even when the reward signals are generated by the same LLM, ICRL prompting still improves performance, highlighting a promising new paradigm for test-time scaling.

## 1 INTRODUCTION

For Large Language Models (LLMs) to act as effective agents on novel tasks, they must be able to improve during inference time, a capability often referred to as test-time scaling (Zhang et al., 2025). Learning and search are the two general methods that can leverage scaling computation for performance improvement (Sutton, 2019), reaching superhuman performance on Chess (Campbell et al., 2002) and Go (Silver et al., 2016). Search has been successfully applied to LLM self-improvement in test-time scaling, starting from the simple Best-of-N (Stiennon et al., 2022) to Tree of Thoughts (Yao et al., 2023) and Monte Carlo Tree Search (Ding et al., 2024).

Learning, however, has yet to receive the same attention for LLM self-improvement at inference time. In-context (supervised) learning (ICL; Brown et al. (2020)), as a supervised learning paradigm, requires expert demonstrations as ground-truth labels. However, such demonstration data are not easily scalable during inference time, which restricts the applicability of ICL to test-time scaling. Thus, LLMs must instead learn from their own generated experience for continual self-improvement (Silver & Sutton, 2024).

---

[*]Equal contribution.

[†]Shangtong Zhang and Yanjun Qi contributed equally as supervising authors. Correspondence to: kefan@email.virginia.edu, amoeini@virginia.edu, shangtong@virginia.edu, yq2h@virginia.edu.

Reinforcement learning is perhaps the most successful algorithm capable of self-improvement independent of human knowledge (Silver et al., 2017). However, its major successes have primarily appeared in simulated environments (Mnih et al., 2015; Silver et al., 2016) or during the training time of LLMs Guo et al. (2025). These current RL settings fall short in the big world setting (Javed & Sutton, 2024), where the real-world environment is vastly more complex than the agent itself. In such environments, agents will encounter numerous situations far beyond their prior training data and must adapt and improve their solutions on the fly. Bridging this gap requires models that (1) can handle diverse tasks in the real world, where natural language often constitutes an essential action space (Silver & Sutton, 2025), and (2) can continually improve their solutions during inference, rather than relying on costly retraining for every novel situation.

This naturally raises the question: can reinforcement learning emerge during the inference phase of LLMs? Enabling LLMs to perform RL purely in context provides an elegant mechanism to meet both requirements: LLM provides a general-purpose initial policy, while RL introduces the capability for continual self-improvement. Inspired by the first surprising evidence that LLMs can act as in-context learners in supervised settings (Brown et al., 2020), a growing body of work has begun to explore in-context reinforcement learning (ICRL; Moeini et al. (2025)). However, current instantiations are largely restricted to bandit or simulated environments (Monea et al., 2025; Krishnamurthy et al., 2024), failing short of addressing many diverse open-ended tasks where natural language is the action space.

In this paper, we bridge this critical gap by demonstrating that LLMs can act as effective in-context reinforcement learners, an emergent capability that improves performance on diverse, language-based tasks ranging from conducting scientific experiments to creative writing to solving olympiad-level mathematics. To reveal this capability, we introduce a simple multi-round prompting framework, **ICRL prompting**. The goal of ICRL prompting is to guide LLMs to perform reinforcement learning for self-improvement on a task. Initially, the prompt is only the task description. After the LLM generates a response, we give numerical scalar feedbacks for the response, called the rewards. Then in the next round, we prompt the LLM again with the same task description and a context consisting of all previous responses and rewards. So on and so forth. We observe that the quality of the LLM's response increases as the context grows. In other words, the LLM is able to maximize the scalar reward signal during the inference time, just like an RL algorithm.

A key design principle of ICRL prompting is minimality. To ensure that the observed gains arise from the emergent RL capacity of LLMs rather than auxiliary mechanisms, we deliberately exclude textual gradients (Yuksekgonul et al., 2025), prioritized experience replay, sampling-based heuristics (Zhang et al., 2024; Yang et al., 2024), or additional engineered modules (Brooks et al., 2024). The only supervision provided is the scalar reward itself. This complies with both the reward hypothesis (Sutton, 2004), "*that all of what we mean by goals and purposes can be well thought of as maximization of the expected value of the cumulative sum of a received scalar signal (reward)*", and the "reward is enough" hypothesis (Silver et al., 2021), "*intelligence, and its associated abilities, can be understood as subserving the maximisation of reward*".

To summarize, this paper makes three contributions:
**(1):** We introduce the *ICRL prompting* framework, a minimal design that elicits inference-time self-improvement in LLMs using only scalar rewards. Just as ICL places $(x, y)$ pairs in context, our framework places state–action–reward tuples with simple meta-instructions in context. This design isolates the LLM's intrinsic capacity for ICRL, free from external code or engineered mechanisms.
**(2):** We provide strong evidence suggesting the emergence of RL in LLM's inference time when the ICRL prompting framework is used. Specifically, we demonstrate the maximisation of the scalar reward signal, the exploration-exploitation trade-off in LLM's inference time, the performance improvements from the growth of the context, the performance drop with short context, and the performance drop when the reward is absent. All those observations are well expected for an RL algorithm. Essentially, this is a "duck test" (Heim, 2007)[1] for the inference process.
**(3):** We demonstrate that ICRL prompting yields significant improvements over self-revision methods such as Self-Refine (Madaan et al., 2023) and Reflexion (Shinn et al., 2023), across diverse benchmarks including Game of 24, creative writing, ScienceWorld, and Olympiad-level mathematics (AIME and HMMT). In Game of 24 and creative writing, the rewards are generated by the LLM itself, yet consistent performance gains are still observed.

---

[1]If it looks like a duck, swims like a duck, and quacks like a duck, then it probably is a duck.

## 2 BACKGROUND

**Reinforcement Learning.** RL uses Markov Decision Processes (MDPs) to model a task, consisting of a state space $\mathcal{S}$, an action space $\mathcal{A}$, a reward function $r : \mathcal{S} \to \mathbb{R}$, an initial distribution $p_0 \in \Delta(\mathcal{S})$ with $\Delta(\mathcal{S})$ denoting the set of probability distributions over $\mathcal{S}$, and a transition function $p : \mathcal{S} \times \mathcal{A} \to \Delta(\mathcal{S})$. At time step 0, an initial state $S_0$ sampled from $p_0$. At time $t$, an agent at $S_t$ takes an action $A_t$ according to its policy $\pi : \mathcal{S} \to \Delta(\mathcal{A})$ with $\Delta(\mathcal{A})$ denoting the set of probability distributions over $\mathcal{A}$, i.e., $A_t \sim \pi(S_t)$. The action $A_t$ is then executed, after which the agent transitions to a successor state $S_{t+1} \sim p(S_t, A_t)$ and recieves a reward $R_{t+1} \doteq r(S_{t+1})$. This agent-environment interaction continues until a time $T$, which marks the end of an episode. The goal of the agent is to adapt its policy $\pi$ such that the expected total rewards $J(\pi) \doteq \mathbb{E}[\sum_{t=1}^{T} R_t]$ is maximized. In modern deep RL (Mnih et al., 2015; Schulman et al., 2017), the policy $\pi$ is usually parameterized by a neural network. We use $\theta$ to denote the network parameter and write the policy as $\pi_\theta$. Typically, RL algorithms update $\theta$ to adapt its policy. For example, at time $t$, the action $A_t$ is sampled from $\pi_{\theta_t}(S_t)$. The RL algorithm then update $\theta_t$ to $\theta_{t+1}$ based on available information such as $S_0, A_0, R_1, \ldots, S_t, A_t, R_{t+1}, S_{t+1}$. Then at time $t+1$, the action $A_{t+1}$ is sampled from the updated policy $\pi_{\theta_{t+1}}(S_{t+1})$. Essentially, the typical RL process is reflected in the updates of $\theta_t$.

**In-Context Reinforcement Learning.** ICRL (Moeini et al., 2025), first coined by Laskin et al. (2023), is an emerging inference-time compute paradigm where the RL process occurs in the inference time (i.e., the forward pass) of the network without any parameter update. In ICRL, the policy $\pi_\theta$ is additionally conditioned on a context called $C_t$, i.e., $A_t \sim \pi_\theta(S_t, C_t)$. The construction of $C_t$ is an active research area but one example is to use all previous state-action-reward pairs obtained in the task. Notably, this usually includes state-action-reward pairs from all previous episodes, not just the current episode (Laskin et al., 2023). In ICRL, there is a pretraining stage where the network $\theta$ is pretrained in a wide range of tasks (MDPs). We use $\theta_*$ to denote the parameter after the pretraining. After the pretraining stage, the policy $\pi_{\theta_*}$ is evaluated in new tasks. In other words, in the new MDP, the action $A_t$ is sampled from $\pi_{\theta_*}(S_t, C_t)$. Importantly, the $\theta_*$ is kept fixed. Nevertheless, it is observed that the quality of $A_t$ increases as $C_t$ grows in the new task. Since $\theta_*$ is fixed, this improvement can only from the increase of the context. This is thus called in-context policy improvement. Notably, this in-context policy improvement is also observed even when the new task is out of the distribution of the pretraining tasks, e.g., Laskin et al. (2023) demonstrate in-context policy improvement in new bandit problems that have the opposite optimal arms to the pretraining bandit problems. Thus this in-context policy improvement cannot be attributed to the hypothesis that $\theta_*$ memorizes the pretraining tasks. The only plausible hypothesis seems to be that the forward pass of the network parameterized by $\theta_*$ implements some RL algorithm to process the information in the context $C_t$ to generate the action $A_t$. This inference-time (forward pass) RL is called in-context RL.

**LLMs as RL Agents.** The token generation process of LLMs can be modeled via RL. In short, the state is all generated tokens and the action is the next token to generate. Namely, let $\mathcal{V}$ be the set of all possible tokens. We consider a state space $\mathcal{S} \doteq \bigcup_{i=1}^{\infty} \mathcal{V}^i$ and an action space $\mathcal{A} \doteq \mathcal{V}$. At time step 0, an initial prompt is given, denoted as $S_0 \in \mathcal{S}$. In this work, $S_0$ contains a description of a task. We refer to the LLM with parameter $\theta$ as $\pi_\theta$. At time $t$, given the current tokens $S_t$, a new token $A_t$ is sampled from $\pi_\theta(S_t)$. The new state is then $S_{t+1} = [S_t A_t]$, i.e., the new state is obtained by concatenating current tokens and the new token. A reward signal $R_{t+1} \doteq r(S_{t+1})$ is then emitted according to a reward function $r$. This token generation process continues until a time $T$, where either $T$ is the maximal allowed response length or $A_{T-1}$ is a special end-of-sequence token. Either way, this marks the end of an episode and the final state $S_T$, called the terminal state, contains both the initial task description and the LLM's response. There are two types of reward functions. One is sparse (the outcome reward model, Ouyang et al. (2022)), where $r(s)$ is nonzero only when $s$ is a terminal state. The other is dense (the progress reward model, Lightman et al. (2023)), where $r(s)$ can also be nonzero for non-terminal states.

## 3 IN-CONTEXT REINFORCEMENT LEARNING PROMPTING

We now present our main contribution, the ICRL prompting framework (Algorithm 1, Figure 1), consisting of the following ingredients.

---

**Algorithm 1** ICRL Prompting

---

**Require:** An LLM $\pi_\theta$. A reward function $r$. Number of episodes $K$. An experience buffer $\mathcal{B}$.
A task description $s_{\text{task}} \in \mathcal{S}$. The ICRL instruction $s_{\text{ICRL}} \in \mathcal{S}$.
1: **for** $k = 1$ **to** $K$ **do**
2:     Construct the initial prompt $S_0$ by concatenating all the tokens in $\mathcal{B}$, $s_{\text{task}}$, and $s_{\text{ICRL}}$.
3:     $t \leftarrow 0$   *// Execute the policy $\pi_\theta$ starting from $S_0$*
4:     **while** $S_t$ is not terminal **do**
5:         $A_t \sim \pi_\theta(S_t), S_{t+1} \doteq [S_t \ A_t], R_{t+1} \doteq r(S_{t+1}), t \leftarrow t + 1$
6:     **end while**
7:     *// $[A_0 \ A_1, \ldots, A_{T-1}]$ is the LLM's response to $s_{task}$ at the current episode*
8:     Push $(A_0, R_1, A_1, R_2, A_2, R_3, \ldots A_{T-1}, R_T)$ into $\mathcal{B}$.
9: **end for**

---

**LLM as the Policy.** An LLM, denoted as $\pi_\theta$, serves as the policy network. The goal is to prompt the LLM to solve a task. We assume a natural language description of the task is available and we denote it as $s_{\text{task}} \in \mathcal{S}$. At the beginning of each episode, we construct the initial prompt by concatenating the LLM's own previous attempts together with the corresponding rewards, the task description, and some meta instruction denoted as $s_{\text{ICRL}}$. The details of the concatenation of previous attempts and the choice of the meta instruction will be discussed shortly. With this initial prompt, the LLM generates the response. Both the response and the rewards are stored in the buffer for future episodes.

**Reward Function.** A numerical scalar reward feedback is provided for each $S_t$ in the episode. Notably, the reward can be either sparse (i.e., only $R_T$ is nonzero) or dense. The reward function can be rule-based, learned separately, or instantiated via the same LLM for self-evaluation. The flexibility of using LLM's self-evaluation as the reward function allows the ICRL prompting framework to be applied to a wide range of tasks. Notably, this scalar reward is the only feedback we provide to the LLM. But we do tell the LLM that this scalar is a reward. We do so by explicitly writing down the word "Reward" before this number when constructing the initial prompt. Notably, if the reward function is rule-based and learned separately, the reward signal constitutes an external feedback. But if the reward function is just the LLM's own evaluation of the answer, there is no external feedback at all in the ICRL prompting framework. Yet we still expect the LLM's response to improve over the episode. This is because of the widely believed hypothesis that evaluation is eaiser than generation. But we do hypothesize that the ceiling with self-evaluation is lower than that with external feedback.

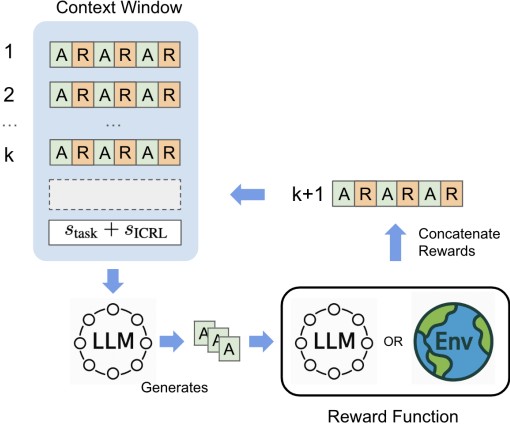

Figure 1: ICRL Prompting. At each episode $k+1$, LLM generates action tokens based on previous experiences up to $k$, and receives numerical rewards either from itself as the evaluator or from the environment. At the end of the episode, the rewards are then concatenated with the action tokens and placed back into the context.

**Memory for Experience.** We use an experience buffer $\mathcal{B}$ to store the LLM's responses and rewards for the task in previous episodes. Our underlying hypothesis is that pretrained LLM already has the ICRL ability. To use this innate ICRL ability to improve LLM's response to the task, we concatenate its previous attempts and rewards as many as the context window allows. We expect that the LLM can reinforcement learn from the experiences in the context during the inference time.

**ICRL Instructions.** To facilitate LLM's inference time RL, we additionally provide some instructions in initial prompt $S_0$ at each episode. The instruction is in natural language and is denoted as $s_{\text{ICRL}}$. We consider three types of instructions: (1) the exploration instruction (Figure 4 in App. A), (2) the exploitation instruction (Figure 5 in App. A), (3) the exploration or exploitation instruction (Figure 6 in App. A). For exploration instruction, the model is asked to provide a response that is

different from all its previous responses. For exploitation instruction, the model is asked to generate the best response based on the the previous responses with the highest reward. We consider two strategies. **(1) ICRL Preset:** We alternate between the exploration and exploitation instructions. When the episode number $K$ is even, we use the exploration instruction. When the episode number $K$ is odd, we use the exploitation instruction. **(2) ICRL Autonomous:** We always provide the "exploration or exploitation" instruction at each episode and let the LLM itself to decide on which to use.

## 4 RELATED WORKS

### 4.1 IN-CONTEXT REINFORCEMENT LEARNING.

The study of inference-time RL algorithms dates back to Duan et al. (2016); Wang et al. (2016), with Laskin et al. (2023) later coining the term in-context reinforcement learning (ICRL), spurring rapid growth in the field (Kirsch et al., 2023; Raparthy et al., 2023; Schmied et al., 2024; Lee et al., 2024; Zisman et al., 2023; Grigsby et al., 2024; Lu et al., 2023; Bauer et al., 2023; Wang et al., 2025; Cook et al., 2024; Xu et al., 2024; Shi et al., 2024; Huang et al., 2024; Liu & Abbeel, 2023; Dai et al., 2024). See Moeini et al. (2025) for a survey. Most existing ICRL works, as a subarea of meta-RL (Beck et al., 2023), use small models trained from scratch in games or robotics. Some employ pretrained LLMs, e.g., as simulators (Brooks et al., 2024; Mirchandani et al., 2023; Resendiz & Klinger, 2025) or in bandit tasks (Krishnamurthy et al., 2024; Nie et al., 2024; Park et al., 2024; Monea et al., 2025), where artificial interventions are often needed and LLMs remain uncompetitive with algorithmic baselines.

### 4.2 INFERENCE-TIME LLM SELF-IMPROVEMENT

Existing methods often rely on natural language self-revision, e.g., Self-Refine (Madaan et al., 2023), Reflexion (Shinn et al., 2023), and Textual Gradient (Yuksekgonul et al., 2025). Since the quality of self-revision is dependent upon the model's parametric knowledge of the task, such approaches are prone to hallucinated feedback that accumulates across iterations, leading to performance collapse (Stechly et al., 2025). In essence, they resemble language-guided search (Liu et al., 2025), where feedback serves as new explicit instructions for the next revision.

By contrast, ICRL requires only numerical rewards, without prescribing new instructions. The model must infer a better response by recognizing patterns from its past experience, making the process akin to reinforcement learning. Crucially, such rewards can also originate directly from the environment, providing a strong source of verification signals.

A parallel line of work improves LLMs at inference via search, e.g., Tree-of-Thoughts (ToT) (Yao et al., 2023), Graph-of-Thoughts (GoT) (Besta et al., 2024), Monte Carlo Tree Search (MCTS) (Ding et al., 2024), and Intelligent Go-Explore (Lu et al., 2025). These methods largely depend on externally engineered components such as heuristics or memory management, rather than leveraging the model's intrinsic learning ability. Our work is also related to previous work on prompt optimization (Yang et al., 2024), where numerical scores guide prompt refinement, though through top-k selection and error filtering. Thus, it is more aligned with in-context supervised learning (e.g., filtered behavior cloning, Grigsby et al. (2024)) than reinforcement learning. In contrast, ICRL enables learning from failure experiences.

## 5 EXPERIMENT

In this section, we evaluate ICRL prompting on three benchmarks: Game of 24, creative writing from Yao et al. (2023), and ScienceWorld (Wang et al., 2022). We compare several baselines including **CoT-only**, **Long-CoT** style prompting, **Best-of-N**, **Self-Refine**, and **Reflexion**. Notably, in all the experiments, we allow the prompt of Self-Refine and Reflexion to grow as long as the LLM allows.

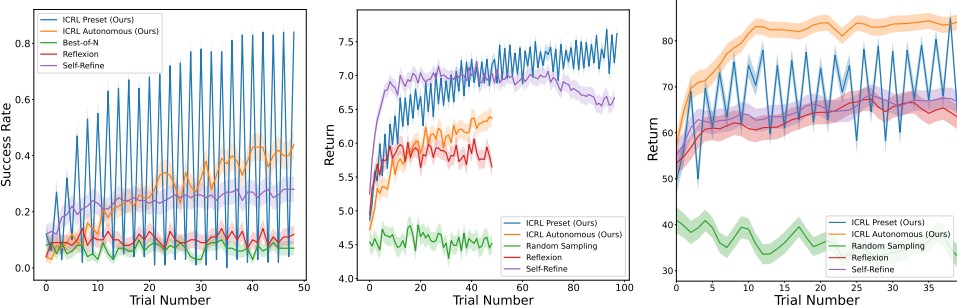

Figure 2: **Baseline Method Comparison. (Left)** Mean Success Rate on Game of 24. **(Middle)** Mean Coherence Reward on Creative Writing. Both ICRL Preset and Self-Refine went through an additional run of 50 episodes. **(Right)** Mean Return on Science World. A running max version of the plots is available in Figure 13 in App. B. This plot shows quality of the response at the current trial while the running max version shows the quality of the best response until now. The shaded region represents $\pm 1$ standard error of the performance calculated across the evaluated tasks.

## 5.1 GAME OF 24

**Task Setup.** Given four input numbers, the model must use each number exactly once and apply only addition, subtraction, multiplication, or division in any order to reach 24. We choose the GPT-4.1 model for this experiment because of its excellent long-context capacity (OpenAI, 2025), accessed through api calls. Following Yao et al. (2023), we use CoT prompting to elicit the model to provide a step-by-step solution, where each step the model picks two numbers from the remaining numbers and an operation to perform the calculation, and obtains one complete solution per LLM query, containing a total of 4 thinking steps. To encure that the LLM generates response with correct format, we additionally provide 5 in-context supervised learning demonstrations. The CoT instruction and the 5 demonstrations together form the task description $s_{\text{task}}$ (Figure 7 in App. A).

**Evaluation.** To verify the correctness of the solutions, we leverage SymPy (Meurer et al., 2017), a library for symbolic mathematics, by extracting operands and operators and evaluating the reconstructed expression to confirm it equals 24, and report the mean success rate over the 100 problems. This rule-based success rate is referred to as $r_*$ (i.e., the ground truth reward function) in the rest of the paper. We use $r$ to denote the reward function that the algorithms actually have access to. In particular, we use GPT-4.1 as the $r$, the same LLM as the policy LLM but prompted differently (see Figure 12 in App. A). After the policy LLM generates the response, for each thinking step, GPT-4.1 scores the likelihood of reaching 24 with the remaining numbers on a 0-3 scale (0 = impossible, 3 = sure). The task in challenging in that no algorithm has access to $r_*$. Instead, they have to rely on their own (possibly imperfect) evaluation, generated by the same LLM, to improve the response.

**Baselines.** We compare our method with CoT-only, Long-CoT style prompting, Best-of-N, Reflexion, and Self-Refine. In CoT-only prompting, the model receives only the task description $s_{\text{task}}$ and produces a single step-by-step solution. In Long-CoT style prompting, we explicitly ask the LLM to generate a long chain-of-thought, and keep retrying if the solution is incorrect in "<think>...</think>" tags, before finally providing the answer. Although GPT-4.1 is not specifically trained for long-form CoT reasoning, we find that Long-CoT style prompting can elicit significantly longer and self-correcting thought traces compared to

Table 1: Game of 24 Success Rate. The running max success rate of the last episode is reported.

| Method | Success Rate |
|---|---|
| CoT-only | 6% |
| Long-CoT | 47% |
| Reflexion | 44% |
| Best-of-N | 49% |
| Self-Refine | 47% |
| **ICRL Preset (Ours)** | **90%** |
| ICRL Autonomous (Ours) | 84% |

zero-shot prompting, making it a strong baseline for the Game of 24. Both methods cannot make

use of a reward signal and are run for one pass. For Best-of-N, to make it even stronger, we use the ground truth reward $r_*$ to select the best response. Self-Refine does not require a reward function. It instead asks the LLM itself to provide textual verbal feedback. Reflexion generates reflection according to $r$. ICRL prompting is different from Self-Refine and Reflexion in that it uses the reward $r$ directly, without any verbal feedback. So the comparision between ICRL prompting and Self-Refine / Reflexion is essentially the comparision between scalar feedback and verbal feedback.

**ICRL prompting.** As discussed before, both $\pi_\theta$ and $r$ in Algorithm 1 are the same LLM GPT-4.1 (prompted differently). We now clarify how we compute $S_0$ at each episode. Since each action is a token, not all actions receive a reward. In fact, since we use CoT to prompt the LLM for 4 thinking steps, only 4 rewards are available for each episode. We thus only include those 4 rewards in $S_0$. We add a "Reward: " tag before the actual scalar reward and then concatenate the tagged reward immediately after the corresponding action (i.e., token).

**Results.** The success rate (i.e., $r_*$) against the number of trials (i.e., the episodes in Algorithm 1) is reported in Figure 2. The ICRL Preset method achieves the highest performance, and the observed oscillations in success rate reflect the model's alternating phases of exploration and exploitation. The mean of running max is also plotted in Figure 13 in App. B. For each problem, we compute its running maximum success rate up to each episode and then average these values across all problems at every episode. As summarized in Table 1, after 50 trials, our methods achieve a success rate of 90% which is significantly higher than 49% from Best-of-N sampling, 47% from Self-Refine, and 44% from Reflexion.

## 5.2 CREATIVE WRITING

**Task Setup.** We consider the creative writing task from Yao et al. (2023), where four sentences are randomly sampled from a pool of sentences. The task for LLMs is to generate four paragraphs, each ending with a sentence, while ensuring that the generated passage is coherent. This is a difficult task, as it challenges the LLMs to craft a unified storyline that logically justifies each of the four sampled sentences by weaving them into a single narrative. A total of 100 problems are evaluated. An example of $s_{\text{task}}$ is in Figure 8 in App. A.

**Evaluation.** We evaluated outputs using the Length-Controlled Alpaca-Eval 2 (Tatsu Lab, 2025) framework, a widely used proxy for human evaluation (Hong et al., 2024; Ethayarajh et al., 2024; Meng et al., 2024) with up to 0.98 Pearson correlation with human judgments. For 100 creative writing problems, we present each method's top response: for Reflexion and Best-of-N, the highest-reward output among 50 trials; for ICRL and Self-Refine, the 50th episode output. Alpaca-Eval then computes pairwise win rates, denoted as $r_*$.

Table 2: Length-Controlled Win Rate (LC) and Standard Error (SE) from Alpaca-Eval 2.0 on Creative Writing.

| Comparison | LC $\pm$ SE (%) |
|---|---|
| Ours vs Reflexion | **59.48** $\pm$ 3.47 |
| Ours vs Long CoT | **78.36** $\pm$ 1.99 |
| Ours vs Self-Refine | **86.32** $\pm$ 3.03 |
| Ours vs Best-of-N | **93.81** $\pm$ 1.01 |

We next introduce the reward function $r$ accessible to the algorithms. We follow standard pairwise comparison (Zheng et al., 2023), using GPT-4.1 with a coherent reference paragraph to score each response from 1–10 (see Fig. 11). Notably, although both $r$ and $r_*$ use an LLM as a judge, they serve distinct purposes. $r$ compares responses against a fixed reference text with emphasis on coherence. By contrast, $r_*$ performs pairwise comparison between two responses generated by our method and a baseline method.

**Baselines.** We compare our method with Best-of-N, Reflexion, and Self-Refine. We allow Best-of-N, Reflexion and ICRL prompting to use $r$. Self-Refine do not use $r$ and instead asks GPT-4.1 to provide verbal feedback. Since it is hard to distinguish CoT and Long-CoT style prompting for this task, we include Long-CoT style prompting as the baseline.

**ICRL prompting.** GPT-4.1 is used as both the policy LLM $\pi_\theta$ and the reward model $r$ in Algorithm 1. At each episode, the initial prompt $S_0$ is constructed by concatenating all of the previous

generations along with their coherence scores from $r$. Notably, this reward is sparse and only $R_T$ can be nonzero. We, therefore, only include $R_T$ in constructing $S_0$.

**Results.** Our method achieves a length-controlled win rate of 59.48% against Reflexion, 78.36 % against Long-CoT style prompting, 86.32 % against Self-Refine, and 93.81 % against Best-of-N as shown in Table 2. This shows the responses generated by our method outperform the ones by baselines in terms of following the instruction to write coherent paragraphs and achieving better human preference. The return curve from reward model $r$ is plotted in Figure 2, and a running max of the return is plotted in Figure 13 in App. B. Although Self-Refine initially matches ICRL in terms of coherence reward, extending both methods by 50 additional episodes, our methods keep improving, whereas self-refine first plateaus, then declines, likely due to the significant growth of its context.

## 5.3 ScienceWorld

**Task Setup.** ScienceWorld (Wang et al., 2022) is an interactive, text-based benchmark consisting of 30 science-experiment tasks set in a multi-room environment populated with diverse objects. The environment is challenging due to sparse rewards, large action spaces, and the requirement for scientific knowledge and efficient exploration. At each step, the agent observes the result of its action and receives zero reward unless it completes a predefined subgoal. This reward signal is used both for evaluation and for inference-time self-improvement (i.e., $r$ and $r_*$ are identical in this task). Completing all subgoals yields a cumulative reward of 100 and marks the episode as successful. An episode ends in failure if the agent either reaches the maximum number of steps or executes an incorrect terminating action. The input $s_{\text{task}}$ provided to the agent describes the environment, the task, and the template of all possible actions. An example of $s_{\text{task}}$ is provided in App. A
**Evaluation.** We use the environment-provided reward function for each task both to construct the trajectories used in the context ($r$), and to evaluate the model ($r^*$). We benchmark each method on all 30 tasks and aggregate the results. GPT-4.1 mini is used as the policy for all compared algorithms.
**Baselines.** In Reflexion, at the end of each episode, the agent is prompted to reflect on its attempt. The reflection is then sanitized and appended to a reflection buffer, which is formatted into the context for subsequent trials. Self-Refine similarly generates self-feedback, but appends it to a trajectory summary, which is then added to the buffer. To ensure a fair comparison, we allow these methods access to the reward signals of the current episode (unlike ICRL) before prompting for reflection.
**ICRL Setup.** Each trial corresponds to a single episode in the environment. After the trial, the new trajectory added to the buffer is constructed by concatenating the actions, observations, rewards, and the final outcome (success or failure). As each episode typically yields only a few rewards, we include only those. At the start of each trial, $S_0$ is constructed by concatenating the task description $S_{\text{task}}$, the collected trajectories, and then the instruction $S_{\text{ICRL}}$. An example of $S_0$ is provided in App.A.
**Results.** The mean return at each trial, is presented in Figure 2 Right. Steady improvements are observed for methods that make use of some form of history of interactions similar to ICRL prompting. However, ICRL prompting outperforms baseline methods by about $20\%$ after enough iterations. To make the comparison fair for efficient baselines such as Best-of-N, in App. B, we compare baslines as we scale test-time compute budget and observe that ICRL also scales better than the baselines not only in terms of number of trials but also the test-time compute budget (in dollar amounts).

## 6 Analysis

**Ablation Study.** To better understand ICRL prompting, we consider following ablations. **(1)** Zero Rewards: We set all rewards to 0. **(2)** Short Context: In Algorithm 1, the buffer $\mathcal{B}$ is essentially a queue of infinite length. Instead, we now make it a deque of length 3. In other words, only the recent 3 episodes are used in constructing $S_0$. **(3)** Exploration Only: We simply ask the LLM to provide a different response than the ones in context, using the exploration instruction as $s_{\text{ICRL}}$, without the reward signal. **(4)** Exploitation Only: We always use the exploitation instruction as $s_{\text{ICRL}}$, with the reward signal. **(5)** No ICRL Instruction: We entirely remove $s_{\text{ICRL}}$.

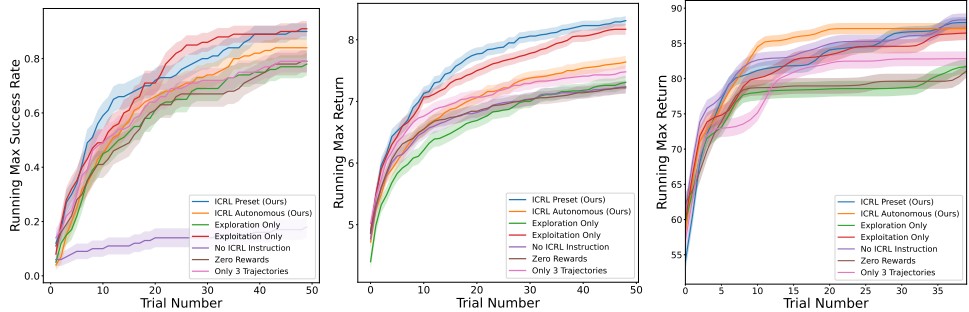

Figure 3: **Ablation Studies (Running Max). (Left)** The mean of running max success rate on Game of 24. **(Middle)** The mean of running max coherence reward on creative criting. **(Right)**. The mean of running max return on ScienceWorld. The shaded region represents $\pm 1$ standard error of the mean (SEM) of the performance calculated across the evaluated tasks within each benchmark.

The running max results of the ablation study are plotted in Figure 3. Both our two methods and the exploitation only with reward signals demonstrate the best-performing curves. This demonstrates our ICRL prompting framework is quite robust to the different prompts setup. We have also observed performance drop with short context and performance drop when the reward is absent. A key finding is that the "exploration only without reward signal" method (shown in green) performs significantly worse than our approach when comparing the maximum performance achieved over time (running max). This demonstrates that our method's improvement is not just due to exploring various responses and then picking the best one previously seen as doing a Best-of-N. Instead, ICRL can genuinely generate novel responses that are better than the ones during the exploration phase.

**Context Length Analysis.** To assess compute efficiency under different input scales, we evaluate ICRL on Qwen3-32B (Yang et al., 2025) across context lengths of 8k, 16k, and 32k (Table 3). The results show that ICRL consistently surpasses Self-Refine and Reflexion in both Creative Writing and AIME, demonstrating superior performance per unit of compute.

**Evaluating ICRL on Open-Source Models and Olympiad-level Mathematics.** To demonstrate the broad applicability of ICRL, we evaluated its performance across two key dimensions: model architecture and task complexity. First, we tested ICRL on a range of open-source models, including Phi-4 (Abdin et al., 2024), Llama-4 Maverick (AI, 2025), Qwen3-32B, and Qwen3-32B-thinking mode (Yang et al., 2025) on the creative writing task. Second, we applied it to challenging Olympiad-level mathematics (AIME (Mathematical Association of America), HMMT (Harvard–MIT

Table 3: Performance of Qwen/Qwen3-32B without reasoning on Creative Writing (LC-WR: length-controlled win rate) and AIME (% solved) across different context lengths.

| Method | CW (LC-WR) | | | AIME (% solved) | | |
|---|---|---|---|---|---|---|
| | 8k | 16k | 32k | 8k | 16k | 32k |
| Self-Refine | 46.33 | 45.83 | 44.41 | 33.33 | **43.33** | 43.33 |
| Reflexion | 42.73 | 40.87 | 40.72 | 30.00 | 30.00 | 33.33 |
| **ICRL** | **50.00** | **50.00** | **50.00** | **40.00** | 43.33 | **46.66** |

Mathematics Tournament)). For details of the experimental setup, please refer to Appendix B.1. ICRL consistently outperforms baselines like self-refine and Reflexion in all settings. Notably, the performance gains are substantial across the board, with improvements of up to 10–20 points over the base model on both creative and mathematical benchmarks. These results demonstrate that ICRL is a robust capability that exists across diverse models and proves effective in challenging task domains.

Table 4: Performance across benchmarks (HMMT, AIME, Creative Writing) for different models and inference-time improvement methods.

| Method | Qwen3 32B (32k) | | | Qwen3 32B think (32k) | | | Llama 4 Maverick (32k) | | | Phi-4 (16k) | | |
|---|---|---|---|---|---|---|---|---|---|---|---|---|
| | HMMT | AIME | CW | HMMT | AIME | CW | HMMT | AIME | CW | HMMT | AIME | CW |
| Base | 9.14 | 22.54 | 34.14 | 52.00 | 66.58 | 1.24 | 8.50 | 17.58 | 0.98 | 5.55 | 20.00 | 10.85 |
| Self-Refine | 16.66 | 43.33 | 46.00 | 56.66 | **83.33** | 30.23 | 13.33 | 20.00 | 45.52 | **13.33** | 33.33 | **51.98** |
| Reflexion | 23.33 | 33.33 | 41.17 | 60.00 | 70.00 | 38.33 | 10.00 | 23.33 | 24.96 | **13.33** | 40.00 | 33.30 |
| **ICRL** | **33.33** | **46.66** | **50.00** | **60.00** | 80.00 | **50.00** | **20.00** | **35.00** | **50.00** | **13.33** | **40.00** | 50.00 |

**Test-Time Learning vs. Test-Time Search.** To verify that ICRL truly learns from external rewards rather than merely searching within the model's parametric knowledge, we evaluated it on generating abstracts for arXiv papers published after the model's training cutoff. In this setting, where the ground truth is absent from the model's training data, standard search methods like Best-of-N and self-correction methods like Reflexion plateau quickly (Figure 17). In contrast, ICRL continues to improve ROUGE-Recall scores over 200 iterations, demonstrating its ability to uncover unseen information solely by exploiting the scalar reward signal. Detailed results are provided in Appendix C.

**Reward-Aware Attention Heads.** To understand how the model internally processes reward signals, we conduct a mechanistic analysis on the creative writing task using Qwen3-32B. Each response in the ICRL trajectory received either a low reward (1) or a high reward (10). For every attention head across the last 32 layers, we computed the mean attention over all response tokens and measured its relationship with the reward using Pearson correlation. We found that many heads consistently track successful examples, placing significantly higher attention on high-reward responses, while other heads track failures, attending more to low-reward responses. This pattern is consistent with classical reinforcement learning where models learn not only from successes but also from failures. Of 2,048 layer–head combinations, 597 (29.1%) show statistically significant correlations, far exceeding the 5% expected by chance. Detailed per-layer results are reported in Appendix C.2.

## 7 CONCLUSION

In this paper, we demonstrate that reinforcement learning is an emergent capability of LLMs at inference time. We show that our minimal, scalar-reward-based ICRL prompting framework unlocks this ability across diverse models and general-purpose tasks, outperforming self-revision methods. A key direction in future work is to investigate how training-time interventions might further enhance this in-context RL capability in LLMs. This surprisingly effective capability points toward a future of more autonomous agents that can explore, adapt, and self-improve in open-ended settings by learning from their own experience.

ACKNOWLEDGMENTS

This work is supported in part by the US National Science Foundation under the awards III-2128019, SLES-2331904, and CAREER-2442098, the Commonwealth Cyber Initiative's Central Virginia Node under the award VV-1Q26-001, a Cisco Faculty Research Award, and an Nvidia academic grant program award.

This material is based upon work supported in part by the National Science Foundation under Grant No. 2124538. Any opinions, findings, and conclusions or recommendations expressed in this material are those of the author(s) and do not necessarily reflect the views of the National Science Foundation.

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

Instruction: Examine all the <attempt>...</attempt> examples, each showing a candidate Response, and the Rewards for each step of the Response. Provide a response that is completely different for any steps from every single one of the previous attempts demonstrated in the context.

Figure 4: The Exploration Instruction ($s_{\text{ICRL}}$).

Instruction: You will be given multiple <attempt>...</attempt> examples, each showing a candidate Response, and the Rewards for each step of the Response. Your task: Based on the previous attempts, try your best to produce a response that can achieve higher rewards.

Figure 5: The Exploitation Instruction ($s_{\text{ICRL}}$).

## A  PROMPT EXAMPLES

Prompt: Write a coherent passage of 4 short paragraphs. The end sentence of each paragraph must be: For some unfathomable reason, the response team didn't consider a lack of milk for my cereal as a proper emergency. You realize you're not alone as you sit in your bedroom massaging your calves after a long day of playing tug-of-war with Grandpa Joe in the hospital. He poured rocks in the dungeon of his mind. I'm a living furnace. Make a plan then write. Your output should be of the following format: Plan: Your plan here. Passage: Your passage here.

Figure 8: An example of $s_{\text{task}}$ for creative writing.

Instruction: Examine all the <attempt>...</attempt> examples, each showing a candidate Response and its Reward. You have two options: exploration or exploitation.

For exploration, provide a response that is completely different for any steps from every single one of the previous attempts demonstrated in the context, while making sure it correctly follows the task instruction.

For exploitation, based on the previous attempts, try your best to produce a response that can achieve higher rewards.

Pick one option to follow.

Figure 6: The Exploration or Exploitation Instruction ($s_{\text{ICRL}}$).

<attempt>
Input: 4 4 6 8
Step1: 4 + 8 = 12 (left: 4 6 12)
Step2: 6 - 4 = 2 (left: 2 12)
Step3: 2 * 12 = 24 (left: 24)
Answer: (6 - 4) * (4 + 8) = 24
</attempt>

<attempt>
Input: 2 9 10 12
Step1: 12 * 2 = 24 (left: 9 10 24)
Step2: 10 - 9 = 1 (left: 1 24)
Step3: 24 * 1 = 24 (left: 24)
Answer: (12 * 2) * (10 - 9) = 24
</attempt>

<attempt>
Input: 4 9 10 13
Step1: 13 - 10 = 3 (left: 3 4 9)
Step2: 9 - 3 = 6 (left: 4 6)
Step3: 4 * 6 = 24 (left: 24)
Answer: 4 * (9 - (13 - 10)) = 24
</attempt>

<attempt>
Input: 1 4 8 8
Step1: 8 / 4 = 2 (left: 1 2 8)
Step2: 1 + 2 = 3 (left: 3 8)
Step3: 3 * 8 = 24 (left: 24)
Answer: (1 + 8 / 4) * 8 = 24
</attempt>

<attempt>
Input: 5 5 5 9
Step1: 5 + 5 = 10 (left: 5 9 10)
Step2: 10 + 5 = 15 (left: 9 15)
Step3: 15 + 9 = 24 (left: 24)
Answer: ((5 + 5) + 5) + 9 = 24
</attempt>

**Task**: Use numbers and basic arithmetic operations (+ - * /) to obtain 24. Put your answer in this format '<answer>**Response** Step1: ... (left: ...) Step2: ... (left: ...) Step3: ... (left: ...) **Answer**: <math operations of the 4 input numbers, even if it does not equal 24></answer>'. Whether it is correct or not, do not try again.
**Prompt**: Input: 1 8 10 11

Figure 7: An example of $s_{\text{task}}$ for Game of 24 with few-shot CoT prompting.

You are a helpful assistant to do some scientific experiment in an environment.
<Environment description> # $s_{task}$
In the environment, there are several rooms: kitchen, foundry, workshop, bathroom, outside, living room, bedroom, greenhouse, art studio, hallway
You should explore the environment and find the items you need to complete the experiment.

The available actions are: Available Actions

FOCUS is a extremely critical action that can be only used the number of times 'focus' is mentioned in the task description. Using it more than that or inappropiately (such as on a wrong object) will terminate the session and the task WILL FAIL.

Task Description:
Your task is to change the state of matter of water. First, focus on the substance. Then, take actions that will cause it to change its state of matter.
</Environment description>

<Instruction> $s_{ICRL}$ </Instruction>

<Attempts> # Buffer

⋮

Attempt $n$:
teleport to bathroom -> Observation: You teleport to the bathroom. (reward=3)
-> focus on substance in toilet -> Observation: You focus on the water. (reward=66)
-> use cup on substance in toilet -> Observation: I'm not sure how to do that. (reward=0)
-> activate sink -> Observation: The sink is now activated. (reward=0)
-> dunk cup into sink -> Observation: No known action matches that input. (reward=0)
-> move cup to sink -> Observation: You move the glass cup to the sink. (reward=0)
-> teleport to kitchen -> Observation: You teleport to the kitchen. (reward=0)
-> use cup on stove -> Observation: I'm not sure how to do that. (reward=0)
-> activate stove -> Observation: The stove is now activated. (reward=2)
-> move cup to table -> Observation: You move the glass cup to the chair. (reward=0)
-> examine cup -> Observation: a glass cup (containing nothing) (reward=0)
Task Failed. You have exceeded the maximum number of steps. (reward=0) Total reward: 71

Attempt $n + 1$:
teleport to bathroom -> Observation: You teleport to the bathroom. (reward=3)
-> focus on substance in toilet -> Observation: You focus on the water. (reward=66)
-> pick up cup -> Observation: You move the glass cup to the inventory. (reward=0)
-> move cup to sink -> Observation: You move the glass cup to the sink. (reward=0)
-> activate sink -> Observation: The sink is now activated. (reward=0)
-> dunk cup into sink -> Observation: No known action matches that input. (reward=0)
-> pour substance in cup into sink -> Observation: You pour the water into the sink. (reward=0)
-> activate stove -> Your generated action "activate stove" cannot be matched to a valid action. (reward=0)
-> teleport to kitchen -> Observation: You teleport to the kitchen. (reward=0)
-> pick up cup containing nothing in table -> Observation: You move the glass cup to the inventory. (reward=0)
-> activate sink -> Observation: The sink is now activated. (reward=0)
Task Failed. You have exceeded the maximum number of steps. (reward=0) Total reward: 69

⋮

</Attempts>

Figure 9: An example of $S_0$ from ScienceWorld.

# Buffer:
**<attempt>**
**Input:** 4 9 10 13.
**Response:**
Step1: 10 - 4 = 6 (left: 6 9 13)     **<Reward**: 3.00>
Step2: 13 - 6 = 7 (left: 7 9)     **<Reward**: 0.00>
Step3: 9 * 7 = 63 (left: 63)     **<Reward**: 0.00>
*Answer**: (13 - (10 - 4)) * 9 = 63     **<Reward**: 3.00>
**</attempt>**

**<attempt>**
**Input:** 4 9 10 13.
**Response:**
Step1: 10 + 4 = 14 (left: 9 13 14)     **<Reward**: 0.00>
Step2: 14 + 9 = 23 (left: 13 23)     **<Reward**: 0.00>
Step3: 23 + 13 = 36 (left: 36)     **<Reward**: 0.00>
*Answer**: (10 + 4 + 9) + 13 = 36     **<Reward**: 0.00>
**</attempt>**

**<attempt>**
**Input:** 4 9 10 13.
**Response:**
Step1: 9 + 10 = 19 (left: 4 13 19)     **<Reward**: 3.00>
Step2: 19 - 13 = 6 (left: 4 6)     **<Reward**: 3.00>
Step3: 6 + 4 = 10 (left: 10)     **<Reward**: 0.00>
*Answer**: ((9 + 10) - 13) + 4 = 10     **<Reward**: 6.00>
**</attempt>**
$s_{\text{ICRL}}$
$s_{\text{task}}$

Figure 10: An example of $S_0$ from Game of 24.

**Instruction:** You are a seasoned text coherence evaluator. Read the TEXT below and rate its overall coherence on a scale from 1 to 10, where 1 means significantly less coherent than the Base Answer, 5 means equally coherent, and 10 means significantly more coherent. Be a strict and conservative evaluator-only assign high scores when the TEXT is clearly better than the Base Answer.

**Base Answer:**
```
{At dawn, golden light slips past pale curtains, rousing the world
in quiet celebration.  A lone robin greets the morning with a
clear, cheerful trill, its song drifting across dew-laden grass.  A
gentle breeze stirs the leaves, carrying the fresh, earthy scent
of new growth.  Nearby, rooftops and empty streets lie poised
between night's calm and the city's stirring pulse, promising
simple comforts like a warm cup of coffee.  In this tranquil pause,
one senses life's renewal and the gentle invitation to greet the
day with hope and gratitude.}
```

**TEXT:** { model_answer}

**Return your answer in exactly this format:** Coherency score: <integer 1–10>.
**Response:**

Figure 11: Prompt for Pairwise Coherence Evaluation for Reward Model $r$.

> **Rule of the Game of 24:** Use all four numbers provided in the input, without repetition, and only basic arithmetic operations (+, −, ×, ÷) to obtain 24. Only three steps are allowed.
>
> Given the following two remaining numbers from a previous step in the Game of 24, the current step is: `{step}`. Evaluate this step.
>
> Examine the numbers shown in each "left: . . ." after the step and reason whether it is still possible to reach 24: • **Sure** → 3 • **Likely** → 1 • **Impossible** → 0
>
> Return the score in the following format: `**Answer**: <integer score>`
>
> **Response:**

Figure 12: Prompt for single-step evaluation used in the reward model $r$ for Game of 24.

## B    ADDITIONAL EXPERIMENTAL RESULTS

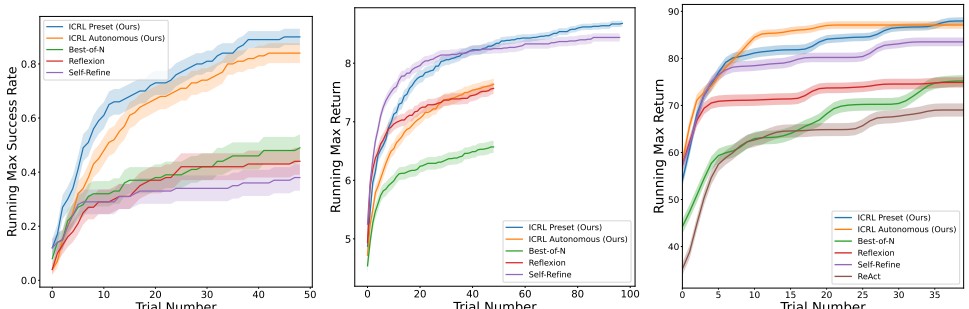

Figure 13: **Benchmark results: Mean of Running Max. (Left)** The mean of running max success rate on Game of 24. **(Middle)** The mean of running max coherence reward on creative writing. Both ICRL Preset and Self-Refine went through an additional run of 50 episodes. **(Right)**. The mean of running max return on ScienceWorld.

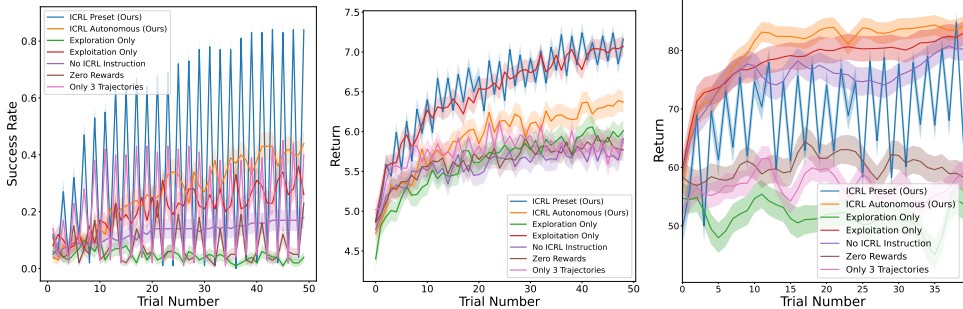

Figure 14: **Ablation study results: Original Curves. (Left)** The mean of success rate on Game of 24 ablation studies. **(Middle)** The mean of coherence reward on creative writing ablation studies. **(Right)**. The mean return on ScienceWorld ablation studies.

Table 5: Running max of return averaged over all the tasks in **ScienceWorld**.

| Method | Return (max = 100) |
|---|---|
| ReAct | $69 \pm 1.4$ |
| Reflexion | $74 \pm 1.1$ |
| Best-of-N | $75 \pm 1.2$ |
| Self-Refine | $83 \pm 0.9$ |
| **ICRL Preset (Ours)** | $\mathbf{88} \pm 0.7$ |
| ICRL Autonomous (Ours) | $87 \pm 0.8$ |

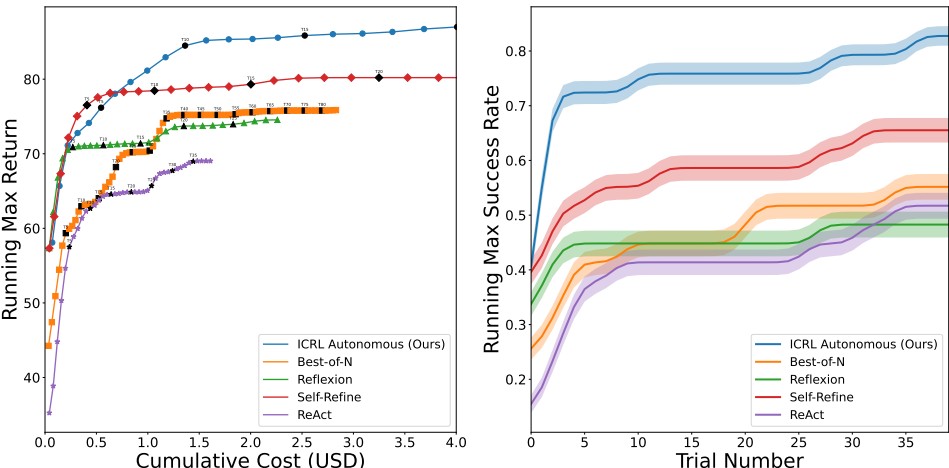

Figure 15: **Additional ScienceWorld Results. (Left)** Although ICRL's context (comprising the experience buffer) is longer than that of random sampling methods, it still outperforms them and other experience-based approaches given additional compute budget. **(Right)** ICRL's superior return improvement as seen in other results, also leads to a greater increase in success rate.

### B.1  ADDITIONAL BENCHMARK: MATH COMPETITIONS

**Task Setup.** AIME (Mathematical Association of America) and HMMT (Harvard–MIT Mathematics Tournament) are two of the most difficult mathematics competitions in the U.S. for high school students. They require not only strong mathematical knowledge but also advanced problem-solving skills. Training language models for reasoning has been one of the most effective methods for improving performance on these benchmarks (DeepSeek-AI et al., 2025).

We aim to test whether reward signals derived from previous reasoning traces can improve subsequent reasoning attempts. To access these traces, we rely on open-source models. In addition to evaluating reasoning models on these benchmarks, we also include a broader set of open-source models to demonstrate the prevalence of ICRL capability across a wide spectrum.

In this benchmark, the model is given a single question and asked to produce an answer. The only additional instruction concerns formatting: the model must place its final answer within `<answer></answer>` tags. The dataset includes the ground-truth answer for each question.

For the reward model, we use the model proposed by Su et al. (2025), which provides a denser signal than simply checking with SymPy whether the model's output matches the ground truth. Using such a model aligns with our belief that the growing ecosystem of judge and reward models, primarily designed for reinforcement fine-tuning of language models (Lightman et al., 2023), will also play a key role in enabling ICRL as a test-time scaling method.

**Evaluation.** To verify correctness, we parse the model's output and check whether its answer matches the ground truth answer provided in the datasets.

**Baselines.** We compare our method with Reflexion and Self-Refine, the two strongest baselines from the other benchmarks. Their implementations are consistent with those used in the prior experiments.

**ICRL Setup.** For attempts after the first, we insert the model's previous answer (including reasoning or CoT tokens) into the context, along with its associated reward. We then prompt the model either to try a new method or to refine its best previous approaches. We truncate past answers to ensure that at least 32 prior attempts can fit in the context.

**Results.** Our method outperforms the baselines in almost all cases. For the reasoning mode of Qwen3-32B, our approach remains competitive with Self-Refine on AIME and surpasses it on HMMT, which is the harder benchmark. Notably, this is achieved using scalar rewards computed with three orders of magnitude less computation, while Reflexion and Self-Refine are allowed nearly twice the compute budget for generating long-CoT verbal feedback.

# C ADDITIONAL ANALYSIS RESULTS

## C.1 UNSEEN PAPER ABSTRACT GENERATION

To isolate whether ICRL truly learns from external rewards rather than merely searching or selecting from the model's parametric knowledge, we design a task where search within parametric knowledge alone is expected to be ineffective.

**Setup.** We fetch 30 arXiv papers published after GPT-4.1-mini's training cutoff. Provided only the title of the papers, the model is instructed to generate the abstract. The goal is to uncover the unseen expert-written abstract as much as possible. We measure performance using ROUGE-recall between the model's generated abstract and the ground-truth abstract, and we use the same score as the external reward. Because these abstracts lie outside the model's pre-training distribution, success requires learning from external rewards rather than searching through memorized content.

**Results.** As shown in Fig. 17, Best-of-1024 sampling reaches only 0.44 ROUGE-recall, indicating that direct search over the model's base distribution cannot recover the missing content. Self-Refine plateaus within a few rounds to 0.45 because it does not use external reward. Reflexion performs slightly better at 0.46 but largely mirrors Self-Refine, suggesting that its revisions are dominated by the model's self-verbal feedback, which carries little useful information in this setup. In contrast, ICRL continues to improve over 200 iterations and achieves substantially higher ROUGE-recall at 0.59, demonstrating that it can effectively learn from the external reward signal and is not limited by the model's pre-training knowledge.

## C.2 REWARD-SENSITIVE ATTENTION HEADS

We conduct a mechanistic analysis to investigate how Qwen3-32B internally processes reward signals during ICRL on the creative writing task.

**Setup.** For each of 100 questions, we select 5 trials from the ICRL trajectory and construct two prompts that differ only in the reward labels: a *test* prompt where each trial's reward is independently randomized to 1 or 10 with equal probability, and a *baseline* prompt where all rewards are set to 1. For every attention head across the last 32 layers (2,048 heads total), we compute the mean attention over all response tokens in each trial, subtract the baseline to isolate the reward-dependent component of attention, and measure its relationship with the reward label using Pearson correlation (500 observations per head: 100 questions $\times$ 5 trials).

**Results.** Figure 16 shows the number of significant heads per layer at $\alpha = 0.05$. Of 2,048 heads tested, 597 (29.1%) are significant, far exceeding the $\sim$102 (5%) expected by chance (dashed line). Many heads consistently track successful examples, placing significantly higher attention on high-reward responses (249 positively correlated heads). Other heads track failures, attending more to low-reward responses (348 negatively correlated heads). This pattern is consistent with classical reinforcement learning where models learn not only from successes but also from failures.

Across layers, reward sensitivity increases toward earlier layers: layers $-27$ to $-32$ average 29.7 significant heads per layer, while layers closest to the output ($-1$ to $-6$) average 12.2. The balance between the two types of heads also varies by depth. In layers closest to the output, most significant heads attend to low-reward responses. In middle layers ($-14$ to $-21$), more heads attend to high-reward responses. The earliest analyzed layers contain a high density of both. These results provide mechanistic evidence that the model actively processes the scalar reward signal across a broad set of attention heads.

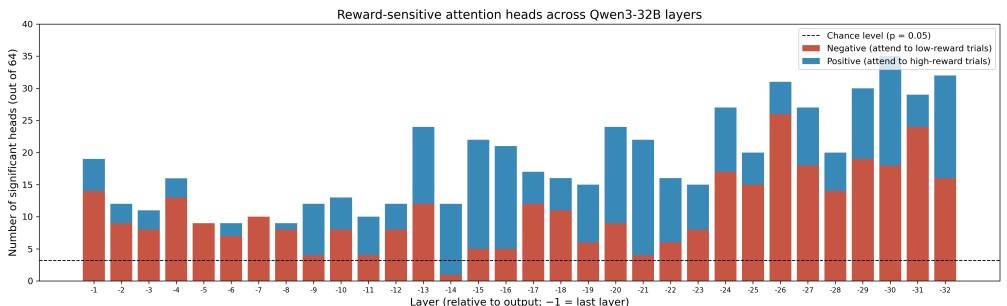

Figure 16: Number of attention heads with statistically significant Pearson correlation ($p < 0.05$) between baseline-adjusted attention and reward label, per layer of Qwen3-32B. Blue: positive correlation (attend more to high-reward trials). Red: negative correlation (attend more to low-reward trials). Dashed line: expected count under chance (3.2 heads per layer).

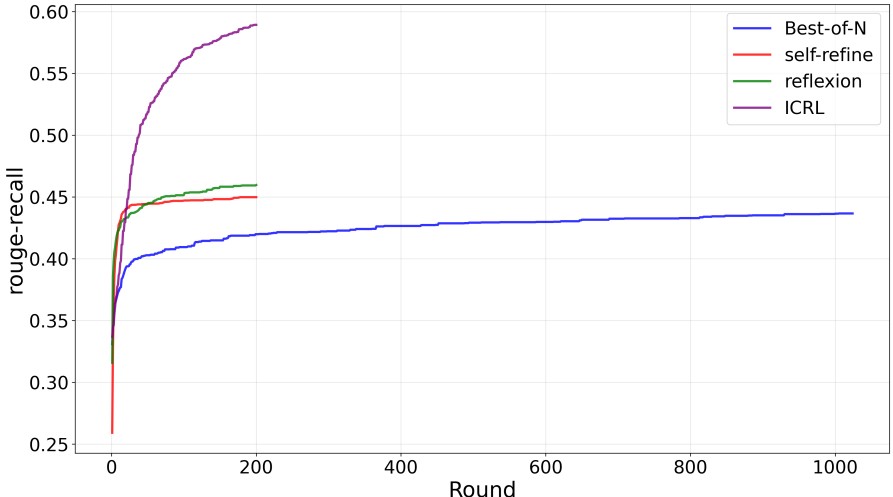

Figure 17: ROUGE-Recall on Generating Unseen Paper Abstracts

