# OpenReview forum: "Reward Is Enough: LLMs Are In-Context Reinforcement Learners"
_ICLR.cc/2026/Conference — ICLR 2026 Poster_

### Official Review · Reviewer_iBtQ · 2025-10-26

**Soundness:** 1
**Presentation:** 2
**Contribution:** 1
**Rating:** 2
**Confidence:** 4

**Summary:**

The authors propose a minimal in-context RL prompting framework where an LLM iteratively generates responses to a task, receives scalar numerical rewards, and conditions future outputs on previous responses and rewards. Over multiple rounds prompting, the model’s behavior exhibits classical RL traits, e.g., reward maximization and context-dependent improvement, demonstrating that reward-driven self-improvement can emerge from prompting.

**Strengths:**

The paper is clearly written, and the method can be adapted to various environments with minimal modifications. Results span arithmetic puzzle solving, open-ended writing, and interactive text environments, suggesting some generality.

**Weaknesses:**

The method is extremely simple and does not fine-tune the model via RL training. It relies only on inference-time prompting with scalar rewards. If the authors could directly fine-tune the parameters of the large language model, just like training a policy network in reinforcement learning, it would increase the complexity of the method and also enhance the novelty of the paper.

The reward provider is the same LLM as the policy, which will amplify bias and risks reward hacking by style, for example, the model may learn to write to please itself rather than to solve the task objectively. I suggest the authors use an additional large model as the reward model.

The “explore” and “exploit” text labels are strong control signal and might explain some gains independent of any learned internal RL mechanism. The ablation shows the method’s reliance on hand-crafted prompt semantics. More neutral or masked forms of reward or context (e.g., numeric tags without the word “Reward”, randomized labels) could test how much the model depends on semantic priors. I suggest removing the "reward" prompt during the prompting process, or placing the reward signal in other positions of prompting to test the performance of this algorithm.

**Questions:**

1. typo: “At time step 0, an initial state S_0 sampled from p_0.” at line 101.
2. In line 110, using θ_t and θ_{t+1} to distinguish the policy network before and after weight updates is inappropriate, because t represents the reinforcement learning time step. If the authors' method does not perform policy updates at every interaction with the environment, then using t and t+1 to make this distinction is not suitable.
3. At line 113, I think it is inappropriate to arbitrarily name the reinforcement learning policy update process (in-weight RL), as this only adds to the burden of understanding and does not substantially improve the method.
4. Since the state transition of the reinforcement learning environment can be constructed by continuously adding prompts, why not directly fine-tune the parameters of the LLM, rather than only using the inference capability of the LLM?
5. Without a strict compute-matched comparison against Best-of-N, ToT/MCTS, or tuned self-revision, it’s hard to isolate where the gains come from. Though, ICRL provides some scaling and context-length analysis, stronger budget-normalized comparisons would be useful.
6. If the environments in the experimental section can be modeled as reinforcement learning environments, why is it necessary to use LLMs to complete these tasks? These tasks are not complex; for example, in the game of 24, I believe that as long as a reward function is set, classical RL methods can also accomplish such tasks.
7. I suggest the authors add credit-assignment analysis to better demonstrate which outputs in the decision sequence the model's performance is strongly correlated with.
8. In Figure 2, the performance variance of ICRL Preset in the game of 24 is very large. Does this indicate that the method is highly unstable? Why does the Self-Refine method have stronger learning efficiency than the method proposed in this paper on Creative Writing? Why does the ICRL Autonomous method only perform excellently on Science World?
9. For creative writing, Alpaca-Eval is a proxy. Pairwise win-rates are useful but should be complemented with human studies or cross-judge robustness to prevent evaluation overfitting. For game of 24, Best-of-N uses a ground-truth solver to pick the best sample, while ICRL uses its own reward r. So, some comparisons are not fully like-for-like across selection rules.
10. I suggest the authors to add Compute-matched comparisons against Best-of-N, ToT/MCTS, and tune Self-Refine/Reflexion under equal token/cost budgets.
11. I suggest the authors to Mask or scramble the “Reward” semantics to test whether numeric tags (without the word “Reward” or with randomized tokens) still work, which will probe whether improvements require semantic understanding vs. mere structure.

**Details Of Ethics Concerns:**

The paper’s core idea is intentionally simple: reuse the model’s own history of attempts and numeric scores to steer the next attempt. It does not apply RL training (no weight updates, no policy gradients); it banks entirely on inference-time behavior and the model’s instruction following. This makes it easy to deploy, at the same time, it also means the “RL emergence” claim remains hypothesis-level, not conclusively distinguished from search/selection + prompting effects.

---

> ### Author Response · Authors · 2025-11-21
>
> We sincerely thank the reviewer for the comprehensive feedback and detailed questions. We appreciate the reviewer's opinion on the generality and flexibility of our method to adapt to various environments. We hope our response below can address the concerns and the questions.
>
>
> > The method is extremely simple and does not fine-tune the model via RL training. It relies only on inference-time prompting with scalar rewards. If the authors could directly fine-tune the parameters of the large language model, just like training a policy network in reinforcement learning, it would increase the complexity of the method and also enhance the novelty of the paper.
>
> Thank you for the insightful comment. While we agree that integrating RL-based fine-tuning is a promising direction for pushing state-of-the-art performance, our work addresses a different fundamental question.
>
> Our primary goal is to determine if LLMs possess intrinsic in-context RL capabilities. Similar to how the original work on in-context learning [1] studied ICL as an emergent property of pre-training rather than fine-tuning, we deliberately focus on a minimal inference-time setup. This allows us to isolate the phenomenon and demonstrate that this capability exists within the model's frozen weights, avoiding the confounding variables introduced by gradient updates.
>
> That said, we agree that our framework is compatible with fine-tuning, similar to other test-time scaling methods such as self-refine and reflexion. We have updated our conclusion to highlight our inference-time approach can serve as a foundation for future fine-tuning methods.
>
>
> [1] Brown, T., et al. (2020). Language Models are Few-Shot Learners. Advances in Neural Information Processing Systems (NeurIPS).
>
>
>
> > The reward provider is the same LLM as the policy, which will amplify bias and risks reward hacking by style, for example, the model may learn to write to please itself rather than to solve the task objectively. I suggest the authors use an additional large model as the reward model.
>
> We appreciate the reviewer’s concern regarding the potential for reward hacking. We would like to clarify that only 2 of our 5 benchmarks use model self-generated proxy rewards, the remaining 3 (AIME, HMMT, and ScienceWorld) use external, rule-based or verifier-based reward functions. For AIME and HMMT specifically, the reward comes from an additional large verifier model that was fine-tuned independently of the policy model. This demonstrates that ICRL is not reliant on self-generated rewards and can learn effectively from external reward sources.
>
> Importantly, we observe consistent improvement not only in the proxy reward signal but also in the ground-truth task metrics. Under reward hacking, proxy reward typically increases while ground-truth accuracy stagnates or declines. Since both metrics improve in our experiments across all tasks, this suggests that the model is learning useful behavior rather than exploiting flaws in the proxy signal.

---

> > ### Author Response · Authors · 2025-11-21
> >
> > > The “explore” and “exploit” text labels are strong control signal and might explain some gains independent of any learned internal RL mechanism. The ablation shows the method’s reliance on hand-crafted prompt semantics. More neutral or masked forms of reward or context (e.g., numeric tags without the word “Reward”, randomized labels) could test how much the model depends on semantic priors. I suggest removing the "reward" prompt during the prompting process, or placing the reward signal in other positions of prompting to test the performance of this algorithm.
> > > I suggest the authors to Mask or scramble the “Reward” semantics to test whether numeric tags (without the word “Reward” or with randomized tokens) still work, which will probe whether improvements require semantic understanding vs. mere structure.
> >
> > We appreciate the reviewer for providing us with actionable suggestions. Before we dive into the additional results, we would like to mention that in one of our ablation study, exploit only also achieves good performance. In this setting, the prompt only instructs the model to produce a response that obtains a higher reward, without mentioning exploration/exploitation strategies (or control signals). This ablation still achieves strong performance, suggesting that the improvements are not solely driven by handcrafted prompt semantics.
> >
> > In light of the reviewer's suggestions, we did the following additional studies with Qwen-3 32B on creative writing and game of 24.
> > 1. We have changed the position of the reward signal from before the response to after the response within an attempt.
> > 2. We have changed the \*\*Reward\*\* label to \*\*Score\*\*.
> > 3. We have removed the word Reward entirely, and changed \*\*Reward\*\* to \*\*\*\*, followed by the numerical reward.
> >
> > | Method                      | CW (% win-ratio vs Self-Refine) | AIME (% solved) |
> > |-----------------------------|----------------------------------|------------------|
> > | ICRL w/ First Position Reward   | 62.80                            | 43.33            |
> > | ICRL w/ Score Label       | 65                            | 45.00           |
> > | ICRL w/ Reward Label Removed   | 75.58                           | 50.00            |
> > | ICRL Preset       | 62.00	                              | 46.66                 |
> >
> > Across all variants, performance remains comparable to the original ICRL Preset. Notably, removing the "Reward" label entirely leads to improved results on both tasks, suggesting that the literal “reward’’ semantics may have introduced unnecessary bias or distraction rather than helping the model’s in-context learning dynamics.
> >
> >
> >
> > > In line 110, using θ_t and θ_{t+1} to distinguish the policy network before and after weight updates is inappropriate, because t represents the reinforcement learning time step. If the authors' method does not perform policy updates at every interaction with the environment, then using t and t+1 to make this distinction is not suitable.
> >
> >
> > Thanks for the feedback! We are describing the typical RL algorithms that requires parametric updates in this section. We have updated the manualscript to emphasize that this is only required for typical RL algorithms that requires parametric updates.
> >
> > > At line 113, I think it is inappropriate to arbitrarily name the reinforcement learning policy update process (in-weight RL), as this only adds to the burden of understanding and does not substantially improve the method.
> >
> > Thank you for pointing this out. The term in-weight RL was borrowed from prior ICRL work [2], but as it is not crucial to understanding our method, and is not used in later sections, we agree it may introduce unnecessary confusion. We have removed it for clarity.
> >
> > [2] Laskin et al., "In-context Reinforcement Learning with Algorithm Distillation," ICLR, 2023.

---

> > > ### Author Response · Authors · 2025-11-21
> > >
> > > > Since the state transition of the reinforcement learning environment can be constructed by continuously adding prompts, why not directly fine-tune the parameters of the LLM, rather than only using the inference capability of the LLM?
> > >
> > > Direct RL fine-tuning is known to suffer from catastrophic forgetting and degradation of general capabilities. Recent state-of-the-art reasoning models such as DeepSeek-R1[3] and agentic models such as GLM-4.5[4], explicitly avoid pure RL fine-tuning in their final models. Instead, they rely on a base LLM plus supervised fine-tuning on a carefully mixed set of RL trajectories and general domain tasks to retain general capabilities. Fine-tuning for every new task would therefore require such costly distillation processes to prevent degradation. In contrast, our inference-only method have demonstrated good performance on several tasks and can be generalized to new tasks with no performance degredation.
> > >
> > >
> > > [3] DeepSeek-AI et al., "DeepSeek-R1: Incentivizing Reasoning Capability in LLMs via Reinforcement Learning," arXiv preprint, 2025.
> > > [4] Zeng et al., "GLM-4.5: Agentic, Reasoning, and Coding (ARC) Foundation Models," arXiv preprint, 2025.
> > >
> > >
> > > > Without a strict compute-matched comparison against Best-of-N, ToT/MCTS, or tuned self-revision, it’s hard to isolate where the gains come from. Though, ICRL provides some scaling and context-length analysis, stronger budget-normalized comparisons would be useful.
> > >
> > > > I suggest the authors to add Compute-matched comparisons against Best-of-N, ToT/MCTS, and tune Self-Refine/Reflexion under equal token/cost budgets.
> > >
> > > Thanks for the suggestion! We calculated the token cost for ScienceWorld and included it in the Appendix B, Figure 15, where our method shows the best cost-effectiveness. We also added a compute-matched comparison on Creative Writing with the GPT-4.1 model. Self-Refine and Reflexion have good cost-adjusted performance when the number of iterations is small (e.g., 10), which is expected since verbal feedback is especially effective in early revisions for writing tasks. However, ICRL outperforms them once the iteration count reaches 50 and 70. ICRL also surpasses Best-of-N even when N becomes very large (e.g., 857). Since Self-Refine does not have a reward design, we additionally evaluated a Best-of-N version, and ICRL still outperforms it, indicating that the improvement is indeed cost-effective.
> > >
> > > | ICRL Iteration | Best-of-N | Best-of-N Iterations | Self-Refine | Reflection | BoN + Self-Refine |
> > > |----------------|------|---------------------|--------------|-------------|---------------------|
> > > | 10             | 88.00 | 25              | 41.52       | 41.69      | 41.65              |
> > > | 30             | 88.93 | 166             | 83.21       | 49.93      | 49.51              |
> > > | 50             | 88.47 | 441            | 83.25       | 56.11      | 55.96              |
> > > | 70             | 85.85 | 857            | 85.64       | 56.15      | 62.51              |
> > >
> > >
> > > > If the environments in the experimental section can be modeled as reinforcement learning environments, why is it necessary to use LLMs to complete these tasks? These tasks are not complex; for example, in the game of 24, I believe that as long as a reward function is set, classical RL methods can also accomplish such tasks.
> > >
> > > We agree that, given a well-shaped reward and sufficient questions to train  on, classical RL could in principle learn Game of 24 from scratch. But we would like to mention that many of our tasks, such as creative writing and Olympiad-style math problems, have natural-language action spaces with extremely large branching factors. The effective search space grows roughly as vocab_size^(answer_length), making RL from a random initial policy impractical. In these settings, an LLM provides the necessary prior to generate meaningful actions to start, and can achieve good performance in fewer iterations.

---

> > > > ### Author Response · Authors · 2025-11-21
> > > >
> > > > > I suggest the authors add credit-assignment analysis to better demonstrate which outputs in the decision sequence the model's performance is strongly correlated with.
> > > >
> > > > We thank the reviewer for the suggestion. We conducted a mechanistic analysis on the creative writing task using Qwen-32B. Each response received either a low reward (1) or a high reward (10). For every attention head in the final layer, we computed the mean attention over all response tokens and measured its relationship with the reward using Pearson correlation and a two-sample t-test.
> > > >
> > > > We found that several heads (e.g., 34, 6, 12) consistently track successful examples, as they place significantly higher attention on high-reward responses and exhibit positive correlation with the reward. Conversely, other heads (e.g., 22, 31) appear to track failures, showing higher attention on low-reward responses and negative correlation with the reward. This pattern is consistent with classical reinforcement learning where models learn not only from successes but also from failures.
> > > >
> > > >
> > > >
> > > >
> > > > ## Attending to Success (High Reward increases attention)
> > > >
> > > > | Head | Pearson r | p-value | Mean Reward=1 | Mean Reward=10 | Sig? |
> > > > | --- | --- | --- | --- | --- | --- |
> > > > | 34 | 0.3058 | 1.41e-04 | -1.6e-05 | +1.8e-05 | ✓ |
> > > > | 6 | 0.2471 | 2.30e-03 | -1.9e-05 | +1.0e-06 | ✓ |
> > > > | 12 | 0.2030 | 1.27e-02 | -6.0e-06 | +1.0e-05 | ✓ |
> > > > | 32 | 0.1867 | 2.21e-02 | -1.5e-05 | +7.0e-06 | ✓ |
> > > > | 29 | 0.1869 | 2.20e-02 | -5.0e-06 | +6.0e-06 | ✓ |
> > > > | 30 | 0.2151 | 8.22e-03 | -1.4e-05 | 0.0e+00 | ✓ |
> > > >
> > > > ## Attending to Failure (Low Reward increases attention)
> > > >
> > > > | Head | Pearson r | p-value | Mean Reward=1 | Mean Reward=10 | Sig? |
> > > > | --- | --- | --- | --- | --- | --- |
> > > > | 22 | -0.2575 | 1.47e-03 | +8.0e-06 | -7.0e-06 | ✓ |
> > > > | 31 | -0.2280 | 5.02e-03 | -3.0e-06 | -1.9e-05 | ✓ |
> > > > | 39 | -0.1982 | 1.50e-02 | +3.0e-06 | -5.0e-06 | ✓ |
> > > > | 37 | -0.1819 | 2.59e-02 | -1.0e-06 | -4.0e-06 | ✓ |
> > > >
> > > > > In Figure 2, the performance variance of ICRL Preset in the game of 24 is very large. Does this indicate that the method is highly unstable?
> > > >
> > > > The variance reflects alternating the explore and exploit rounds for ICRL preset. In the explore rounds the average performance drops, whereas in exploit rounds it raises, producing a predictable periodic oscillation, rather than algorithmic instability.
> > > >
> > > > > Why does the Self-Refine method have stronger learning efficiency than the method proposed in this paper on Creative Writing?
> > > >
> > > >
> > > > The task of creative writing primarily tests the model’s ability to write a coherent passage based on four random sentences. This is exactly the type of task where verbal feedback and self-revision methods tend to perform best, since instruction-tuned LLMs can directly identify weakness in coherency and suggests revisions. In this light, it is expected that Self-Refine shows higher early-stage efficiency. What is more suprising is that ICRL, despite relying only on a scalar reward, can eventually match or surpass it.
> > > >
> > > >
> > > > > Why does the ICRL Autonomous method only perform excellently on Science World?
> > > >
> > > > We expected ScienceWorld to exhibit different behavior because it is an agentic benchmark with many steps and states. In this setting, the model has a better opportunity to infer whether it should explore or exploit, compared to non-agentic (e.g. math or creative writing) tasks where this decision is less clear.
> > > >
> > > >
> > > > > For creative writing, Alpaca-Eval is a proxy. Pairwise win-rates are useful but should be complemented with human studies or cross-judge robustness to prevent evaluation overfitting. For game of 24, Best-of-N uses a ground-truth solver to pick the best sample, while ICRL uses its own reward r. So, some comparisons are not fully like-for-like across selection rules.
> > > >
> > > >
> > > >
> > > > For game of 24, we were prompting the LLM itself as the reward model, which is noisy and probabalistic. To avoid underestimating best-of-n, we utilized the deterministic ground-truth verifier for the selection step. This reveals the best-case performance of the Best-of-N and ensures our method is compared against a strong and stable baseline.

---

> ### Author Response · Authors · 2025-11-21
>
> > the “RL emergence” claim remains hypothesis-level, not conclusively distinguished from search/selection + prompting effects.
>
> We thank the reviewer for raising this important concern. To test whether our method is merely a better search/selection procedure, we designed a task where search over prior knowledge should fail. (For more details and plots, please refer to the updated Appendix C.)
>
> Specifically, we ask the model to write an abstract given only a paper title, using 30 arXiv papers published after the model’s training cutoff, and evaluate how much the model’s response can recover the ground-truth abstract by ROUGE-recall.
>
> In this setting, even Best-of-1024 sampling achieves ROUGE-recall around 0.44 against the ground-truth abstracts, indicating that “just sample more” does not recover the unseen content. In the same setup, self-refine quickly plateaus at 0.45, and Reflexion offers only minor gains at 0.46, suggesting that, similar to self-refine, its revisions are dominated by the model’s self-verbal feedback, which carries little useful information in this setup.
>
> By contrast, ICRL continues to improve over iterations and clearly outperforms these test-time scaling baselines on this task at 0.59. Since the target texts are out-of-distribution with respect to the model’s pre-training data, this performance gap is difficult to explain as mere search over parameteric knowledge. Instead, it indicates that ICRL can make more effective use of external rewards at test time.

---

### Official Review · Reviewer_nrTz · 2025-10-30

**Soundness:** 4
**Presentation:** 3
**Contribution:** 2
**Rating:** 6
**Confidence:** 3

**Summary:**

The paper proposes an in-context RL framework that provides a task description, and then judges the answers of the model to provide a reward that is used by the LLM to improve its answers. The authors evaluate the method on different text-based RL problems, where the method outperforms other self-revision techniques like Self-Refine and Reflexion.

**Strengths:**

The paper is well-written and easy to follow. The method seems sound and is able to achieve good performance. There is good diversity in the tasks that the method is evaluated on, ranging from math puzzles to creative writing.

**Weaknesses:**

I am unsure about the novelty of the method. It seems that a number of works on ICRL already exist in the literature. The authors state that existing works are "restricted to bandit or simulated environments [...], falling short of addressing complex open-ended tasks in the real world", which seems a bit vague. There are other papers that do in-context reinforcement learning that also tackle complex sequential decision-making tasks, e.g., [1-4]. One of the claimed contributions is the "ICRL prompting framework", but it is unclear how this framework differs from existing frameworks used in ICRL. I would greatly appreciate it if the authors could elaborate on the exact novelty in their method and setting.

[1] Michael Laskin et al. "In-context Reinforcement Learning with Algorithm Distillation." The Eleventh International Conference on Learning Representations.
[2] Yarik Menchaca Resendiz, and Roman Klinger. "PARL: Prompt-based Agents for Reinforcement Learning." arXiv preprint arXiv:2510.21306 (2025).
[3] Ethan Brooks et al. "Large Language Models can Implement Policy Iteration." Advances in Neural Information Processing Systems 36 (2023).
[4] Sili Huang et al. "In-context decision transformer: Reinforcement learning via hierarchical chain-of-thought." arXiv preprint arXiv:2405.20692 (2024).

**Questions:**

1. Figure 2/3 middle + right: The intuition of the returns is not clear, which makes it hard to assess the performance level and the significance of the performance differences between the different configurations. Is the agent actually able to solve the task? What is the difference between two agents that have X difference in return? Are both solving the tasks more or less, or is there a perceptible difference in the quality of the solutions? Maybe there is a more intuitive metric for these tasks than the return that would make it easier to assess the performance of the agents.

2. Figures 2 and 3: What exactly do the shaded areas represent?

3. Line 370 mentions a "length-controlled win rate": How is that defined?

4. Figure 3: It is surprising that the "zero rewards" configuration actually improves and even reaches somewhat decent performance. Perhaps the authors could elaborate on how the agent is able to improve without any kind of feedback.

5. Figure 3 middle: Some of the curves seem incomplete. ICRL Autonomous is still improving, so it is unclear whether it would eventually reach the performance of ICRL Preset and Self-Refine. Is there a reason why some curves only go to trial 50 here?

Typos:

Line 124: "can only from": word missing

Line 139: "funciton" --> "function"

Line 298: "The task in challenging"

Line 444: "criting" --> "writing"

---

> ### Author Response · Authors · 2025-11-21
>
> We deeply thank the reviewer for reviewing our work and acknowledging the soundness of our method, and the diversity of the tasks evaluated. We hope our response below addresses the reviewer's concerns and questions.
>
> > The authors state that existing works are "restricted to bandit or simulated environments […], falling short of addressing complex open-ended tasks in the real world", which seems a bit vague.
>
> We thank the reviewer for helping us improve the precision of our claims. To clarify that our focus is specifically on the complexity of language as the action space rather than physical environments, we revised the description from “falling short of addressing complex open-ended tasks in the real world” to the more precise “failing to address many diverse open-ended tasks where natural language is the action space.” We also changed “general-purpose tasks in the real world” to “diverse tasks in the real world, where natural language often constitutes an essential action space” to emphasize that our contribution concerns settings where the action space is natural language.
>
> > unsure about the novelty of the method.
>
> Our work shows that LLMs can perform in-context reinforcement learning directly from reward signals, which is a capability that, to our knowledge, has not been demonstrated for diverse natural-language tasks. For the test-time scaling community, this introduces a new method complementary to existing search-based or verbal-feedback approaches. For the RL community, although ICRL has been studied in the RL tasks, to improve in language modeling and language agent tasks requires RL fine-tuning like PPO and GRPO. Our results demonstrate that ICRL can be leveraged by LLM to self-improve on diverse and complex language tasks, expending the scope of applications.
>
> > There are other papers that do in-context reinforcement learning that also tackle complex sequential decision-making tasks.
>
> Thanks for pointing out these related papers! These works indeed study in-context RL, but their settings differ substantially from ours: they [1, 2, 3, 4] focus on standard RL benchmarks such as chain, distractor-chain, mazes, MiniCatch, MiniInvaders, point-mass control, GridWorld, or MuJoCo tasks (e.g., HalfCheetah, Hopper, Walker). In contrast, our work investigates language modeling and language agent tasks, where the action space is natural language.
>
> Among the cited works, [2] and [3] also use LLMs as base models. However, [3] incorporates known ICL mechanisms as components to build an RL system, while our goal is to test whether LLMs themselves exhibit ICRL capabilities. Meanwhile, [2] focuses on standard RL control tasks we have updated the related-work section to include them.
>
> > One of the claimed contributions is the "ICRL prompting framework", but it is unclear how this framework differs from existing frameworks used in ICRL.
>
> Our goal is to examine whether LLMs can perform ICRL at inference time. Accordingly, our prompting framework is designed directly around this research question: we aim to present the ICRL problem to the model while explicitly avoiding any solutions or algorithmic hints as in prior work such as [3]. To achieve this, our framework includes only the state–action–reward tuples and a minimal meta-prompt. In the exploit-only ablation, this meta-prompt is further reduced to a single description that the task of the LLM is to provide a response to maximize reward.

---

> > ### Author Response · Authors · 2025-11-21
> >
> > > Figure 2/3 middle + right: The intuition of the returns is not clear, which makes it hard to assess the performance level and the significance of the performance differences between the different configurations. Is the agent actually able to solve the task? What is the difference between two agents that have X difference in return? Are both solving the tasks more or less, or is there a perceptible difference in the quality of the solutions? Maybe there is a more intuitive metric for these tasks than the return that would make it easier to assess the performance of the agents.
> >
> > In scienceWorld there are several subtasks in each task the completion of which rewards the agent with points. The sum of the reward of all subtasks is 100. The reward of each subtask is correlated with its difficulty. Consider this simple task: freeze water. Finding a container is 20 points. Filling a container with water has 40 points. Putting the container in the freezer is 40 points. So if a method achieves 10 reward more than another, on average, then it means it completes more subtasks corresponding to 10% of the whole task, on average.
> >
> > > Figures 2 and 3: What exactly do the shaded areas represent?
> >
> > We thank the reviewer for pointing out this ambiguity. We clarify that the shaded region represents $\pm 1$ standard error of the mean (SEM) of the performance calculated across the evaluated tasks. We have updated the captions for Figures 2 and 3 in the revision to explicitly define this statistic.
> >
> > > Line 370 mentions a "length-controlled win rate": How is that defined?
> >
> > We report this metric introduced in the AlpacaEval 2.0 paper [5]. As LLM-based judges often exhibit a bias toward longer responses, the authors developed the 'length-controlled win rate' to statistically adjust for output length. This results in a more robust metric that evaluates model quality with significantly reduced length bias.
> >
> >
> >
> > [5] Dubois et al., "Length-Controlled AlpacaEval: A Simple Way to Debias Automatic Evaluators," COLM, 2024.
> >
> > > Figure 3: It is surprising that the "zero rewards" configuration actually improves and even reaches somewhat decent performance. Perhaps the authors could elaborate on how the agent is able to improve without any kind of feedback.
> >
> > We appreciate this observation. The 'zero rewards' configuration effectively functions as the exploration-only configuration, where the model has all previous attempts in context and generates a new response. This configuration achieves decent performance because it essentially performs a 'search and select' process, which is an effective mechanism in test-time scaling research. However, the wide margin of performance differences confirms that ICRL is not merely relying on simple search and selection, but is successfully utilizing reward signals to guide improvement.
> >
> > > Figure 3 middle: Some of the curves seem incomplete. ICRL Autonomous is still improving, so it is unclear whether it would eventually reach the performance of ICRL Preset and Self-Refine. Is there a reason why some curves only go to trial 50 here?
> >
> > For these experiments, we used the closed-source GPT-4.1 model. We ran all methods up to 50 trials, where ICRL Preset was already outperforming Self-Refine. To see the scaling property of our method against the best performing baseline self-refine on this task, we run additional 50 trials. Due to the API cost, we only run ICRL preset against self-refine. We agree that ICRL Autonomous shows a promising upward trend and would likely improve further with more trials.

---

### Official Review · Reviewer_J3p5 · 2025-10-31

**Soundness:** 3
**Presentation:** 4
**Contribution:** 3
**Rating:** 6
**Confidence:** 5

**Summary:**

This paper introduces a multi-round prompting framework, ICRL Prompting, to argue that LLMs exhibit an emergent capability for in-context reinforcement learning (ICRL). The method appends a history of the LLM's prior (response, scalar reward) pairs to the context. The authors demonstrate that this minimal mechanism, which relies solely on scalar feedback, achieves performance improvements on several benchmarks and outperforms baseline methods that rely on textual feedback.

**Strengths:**

The primary strength of this paper is its novelty. It reveals an interesting phenomenon: simple scalar reward signals within the context may be sufficient to drive self-improvement in LLMs. This claim is supported by impressive empirical results, where the method consistently outperforms strong baselines like Self-Refine and Reflexion on several benchmarks. The experimental design includes rigorous and fair baseline comparisons, notably by allowing baselines like Self-Refine to use an equally growing context and by comparing ICRL against an overpowered Best-of-N baseline that uses the ground-truth reward for selection. The inclusion of a computational cost analysis in the appendix further strengthens the paper's claims of practical utility.

**Weaknesses:**

The paper's most significant weakness is a methodological flaw in its core experiments: a self-referential bias. In several key benchmarks, the reward signal r is generated by the LLM itself. This introduces a high risk of reward hacking, where the policy model may simply be learning to overfit the biases of its own evaluation model rather than optimizing the true task metric r*. The paper fails to provide sufficient evidence to rule out this possibility.

This methodological issue is compounded by the paper's reliance on phenomenological observation (a "duck test") for its central claim that LLMs are RL learners. The paper offers no mechanistic analysis explaining how the Transformer architecture actually utilizes the (response, reward) history to achieve policy improvement.

In addition to this lack of mechanistic depth, the paper does not include a systematic robustness analysis of reward signal quality. The ablation study tests zero rewards, but this is not equivalent to testing the method's tolerance for noisy or incorrect rewards, which is critical for real-world application. This limited scope of evidence leads to the final weakness: the paper's conclusion is overclaimed. All empirical evidence is derived from text-based environments, making the extension of its claims to general-purpose tasks unsupported.

**Questions:**

1. The ablation study tests zero rewards but not noisy rewards. Could the authors provide a sensitivity analysis showing how ICRL's performance degrades if 20% or 50% of the scalar rewards in the context are intentionally replaced with incorrect values?

2. The claim of emergent RL capability is currently based on external behavior. To provide mechanistic evidence, could the authors provide a preliminary interpretability analysis (e.g., attention maps) to show whether the model systematically attends more to tokens associated with high-reward history when generating a new response?

3. The performance gap between ICRL (90% on Game of 24) and the Best-of-N baseline (49%), which used the true reward r*, is striking. What is the authors' intuition for this gap? Does it imply that correct solutions are so sparse in the model's base distribution that simple sampling even with a perfect r* selector fails to find them, whereas ICRL is actively learning to generate them?

4. Could the authors please clarify the context truncation strategy mentioned in Appendix B.1 (used to fit "at least 32 prior attempts")? Is this a simple FIFO queue, or a more complex strategy based on reward (e.g., prioritizing high-reward experiences)?

---

> ### Author Response · Authors · 2025-11-21
>
> We sincerely thank the reviewer for the constructive feedback and detailed questions. We appreciate the reviewer’s acknowledgement of the paper’s novelty on revealing an interesting phenomenon, the rigor of our baseline comparisons, and the practical value of the method. We hope our response below addresses the reviewer's concerns and questions.
>
>
> >a methodological flaw in its core experiments: a self-referential bias
>
> We appreciate the reviewer raising this critical question regarding self-referential bias. We wholeheartedly agree this is a serious methodological flaw to avoid. We confirm that our evaluation benchmarks do not rely on a model-generated self-judgment. We use external, deterministic verifiers for correctness in Game of 24, AIME, and HMMT, a length-bias-adjusted Alpaca-Eval 2.0 with 98% human correlation for Creative Writing, and a rule-based function for ScienceWorld task completion, ensuring that the final "correct" or "complete" state leaves no room for self-referential bias in performance evaluation.
>
> >In several key benchmarks, the reward signal r is generated by the LLM itself. This introduces a high risk of reward hacking
>
> We appreciate the reviewer’s concern regarding the potential for reward hacking. We would like to clarify that only 2 of our 5 benchmarks use model-generated proxy rewards, the remaining 3 (AIME, HMMT, and ScienceWorld) use external, rule-based or fine-tuned verifier-based reward functions. This demonstrates that ICRL is not reliant on self-generated rewards and can learn effectively from external reward sources.
>
> Importantly, we observe consistent improvement not only in the proxy reward signal but also in the ground-truth task metrics. Under reward hacking, proxy reward typically increases while ground-truth accuracy stagnates or declines. Since both metrics improve in our experiments across all tasks, this suggests that the model is learning useful behavior rather than exploiting flaws in the proxy signal.
>
> > This methodological issue is compounded by the paper's reliance on phenomenological observation (a "duck test") for its central claim that LLMs are RL learners. The paper offers no mechanistic analysis explaining how the Transformer architecture actually utilizes the (response, reward) history to achieve policy improvement.
> > The claim of emergent RL capability is currently based on external behavior. To provide mechanistic evidence, could the authors provide a preliminary interpretability analysis (e.g., attention maps) to show whether the model systematically attends more to tokens associated with high-reward history when generating a new response?
>
> We thank the reviewer for this actionable suggestion! We conducted a mechanistic anlaysis on the creative writing task using Qwen-32B. Each response received either a low reward (1) or a high reward (10). For every attention head in the final layer, we computed the mean attention over all response tokens and measured its relationship with the reward using Pearson correlation and a two-sample t-test.
>
> We found that several heads (e.g., 34, 6, 12) consistently track successful examples, as they place significantly higher attention on high-reward responses and exhibit positive correlation with the reward. Conversely, other heads (e.g., 22, 31) appear to track failures, showing higher attention on low-reward responses and negative correlation with the reward. This pattern is consistent with classical reinforcement learning where models learn not only from successes but also from failures.
>
>
>
>
> ## Attending to Success (High Reward increases attention)
>
> | Head | Pearson r | p-value | Mean Reward=1 | Mean Reward=10 | Sig? |
> | --- | --- | --- | --- | --- | --- |
> | 34 | 0.3058 | 1.41e-04 | -1.6e-05 | +1.8e-05 | ✓ |
> | 6 | 0.2471 | 2.30e-03 | -1.9e-05 | +1.0e-06 | ✓ |
> | 12 | 0.2030 | 1.27e-02 | -6.0e-06 | +1.0e-05 | ✓ |
> | 32 | 0.1867 | 2.21e-02 | -1.5e-05 | +7.0e-06 | ✓ |
> | 29 | 0.1869 | 2.20e-02 | -5.0e-06 | +6.0e-06 | ✓ |
> | 30 | 0.2151 | 8.22e-03 | -1.4e-05 | 0.0e+00 | ✓ |
>
> ## Attending to Failure (Low Reward increases attention)
>
> | Head | Pearson r | p-value | Mean Reward=1 | Mean Reward=10 | Sig? |
> | --- | --- | --- | --- | --- | --- |
> | 22 | -0.2575 | 1.47e-03 | +8.0e-06 | -7.0e-06 | ✓ |
> | 31 | -0.2280 | 5.02e-03 | -3.0e-06 | -1.9e-05 | ✓ |
> | 39 | -0.1982 | 1.50e-02 | +3.0e-06 | -5.0e-06 | ✓ |
> | 37 | -0.1819 | 2.59e-02 | -1.0e-06 | -4.0e-06 | ✓ |

---

> ### Author Response · Authors · 2025-11-21
>
> > The ablation study tests zero rewards but not noisy rewards. Could the authors provide a sensitivity analysis showing how ICRL's performance degrades if 20% or 50% of the scalar rewards in the context are intentionally replaced with incorrect values?
>
> We have provided the following results on different levels of noisy rewards. We have observed that when the noise level is 20% and 50%, the performance is on par with Preset, whereas 100% of the reward are assigned as random, the performance drops. This suggests good robustness of ICRL method against noisy rewards.
>
> | Method                               | CW (% win-ratio vs Self-Refine) | AIME (% solved) |
> |--------------------------------------|----------------------------------|------------------|
> | ICRL w/ 20% Random Rewards           | 60.70                           | 46.66           |
> | ICRL w/ 50% Random Rewards           | 73.22                          | 46.66          |
> | ICRL w/ 100% Random Rewards          | 43.11                           | 33.33           |
> | ICRL Preset                                | 62.00                           | 46.66           |
>
>
> > the paper's conclusion is overclaimed. All empirical evidence is derived from text-based environments, making the extension of its claims to general-purpose tasks unsupported.
>
>
> We thank the reviewer for this insightful observation. We agree that our empirical evidence is rooted in natural language tasks and have revised the manuscript to ensure our claims are strictly scoped to this context.
>
> Specifically, we have modified Lines 51-53 replace broad claims of "general-purpose tasks" with specific references to "diverse tasks where natural language constitutes an essential action space.", and Lines  "complex, general-purpose tasks" to "diverse, language-based tasks". We believe this accurately reflects the scope of our contribution while maintaining the significance of the findings.
>
>
> > The performance gap between ICRL (90% on Game of 24) and the Best-of-N baseline (49%), which used the true reward r*, is striking. What is the authors' intuition for this gap? Does it imply that correct solutions are so sparse in the model's base distribution that simple sampling even with a perfect r* selector fails to find them, whereas ICRL is actively learning to generate them?
>
>
> We agree with the reviewer’s assessment. The performance gap highlights a fundamental difference between methods that rely on passive versus adaptive proposal distributions.
>
> Best-of-N relies on passive rejection sampling from a fixed (and often suboptimal) proposal distribution of responses, in this case, the model's base distribution. It is statistically unlikely to sample a correct reasoning chain when it is sparse in the model's base distribution even with a perfect verifier. In contrast, ICRL actively leverages previous experiences in context to shifts the proposal distribution toward the high-reward responses.
>
> > Could the authors please clarify the context truncation strategy mentioned in Appendix B.1 (used to fit "at least 32 prior attempts")? Is this a simple FIFO queue, or a more complex strategy based on reward (e.g., prioritizing high-reward experiences)?
>
> We confirm that the truncation strategy is a simple FIFO queue, maintaining a sliding window of the most recent interactions.

---

> > ### Comment · Reviewer_J3p5 · 2025-11-27
> >
> > I thank the authors for their detailed response and the additional experiments, particularly the mechanistic analysis and noise sensitivity tests. These have successfully addressed my initial concerns regarding self-referential bias and robustness.
> > However, I still have two remaining concerns regarding the theoretical interpretation of the method:
> > The new ablation study showing that the method works equally well with the "Reward" label removed or replaced suggests that the underlying mechanism is closer to numerical pattern matching (attending to tokens associated with high numbers) rather than a semantic understanding.
> > As discussed regarding the zero rewards setting, the framework relies entirely on passive stochastic sampling for exploration. It lacks the active exploration mechanisms that are fundamental to rigorous RL algorithms, which may limit its ability to escape local optima.
> > Therefore, I will maintain my score.

---

> ### Author Response · Authors · 2025-12-03
>
> We are glad that our responses addressed the initial concerns regarding self-referential bias, mechanistic interpretability, and robustness to the reward signal. We hope the following clarifications address the additional concerns.
>
> > The new ablation study showing that the method works equally well with the "Reward" label removed or replaced suggests that the underlying mechanism is closer to numerical pattern matching (attending to tokens associated with high numbers) rather than a semantic understanding.
>
> We thank the reviewer for this insightful intuition. We agree with this interpretation: the ablation results confirm that our method relies on numerical signals rather than the semantics of the "Reward" label itself.
>
> We believe this effectively highlights the distinction between ICRL and self-revision methods (e.g., Reflexion, Self-Refine). Self-revision relies on explicit semantic instructions to guide improvement, but for ICRL, the model can only rely on numerical signals. This requires the model to learn from its past experiences by identifying patterns from both success and failure responses. Importantly, our attention analysis also confirms the model attends to responses associated with both low and high rewards.
>
> > As discussed regarding the zero rewards setting, the framework relies entirely on passive stochastic sampling for exploration. It lacks the active exploration mechanisms that are fundamental to rigorous RL algorithms, which may limit its ability to escape local optima.
>
> We would like note that reliance on stochastic sampling is common in rigorous RL. Algorithms like REINFORCE and PPO also depend on passive sampling from the policy distribution to explore. Our exploitation-only ablation is closest to this setup. We would also like to mention that in our ICRL-Preset, we implement an explicit exploration schedule (50% of rounds, analogous to epsilon-greedy), rather than relying on the purely "passive" behavior from our exploitation-only ablation. We believe that developing more sophisticated active exploration mechanisms in LLMs for ICRL is a promising direction for future research.

---

### Author Response · Authors · 2025-12-03
**Additional Analysis**

We sincerely thank all reviewers for their thoughtful and constructive feedback!

We would like to highlight an additional analysis result in Appendix C, motivated by Reviewer iBtQ's feedback on distinguising from search/select, where we demonstrate that ICRL succeeds in scenarios where the answer falls outside the model's parametric knowledge. In these cases, current test-time scaling methods fail (even after 1,024 samples) as they primarily rely on searching within the model's existing parametric knowledge, whereas ICRL can effectively learn from external rewards to uncover unseen information. We believe this finding demonstrates a fundamental advantage of guiding LLMs to perform ICRL.

---

### Meta-Review · Area_Chair_8zYr · 2026-01-05

**Summary:**

The work investigates how LLMs can improve at test-time by introducing a simple strategy that consist of appending the score to a past attempt at the task, and adding them into the LLM's context. The authors brand this as In-Context RL as they interpret the score as a reward in the RL framework. This simple method leads to consistent gains across well established in-context baselines such as Reflexion. These conclusions are surprising, but seem to be corroborated by additional experiments and qualitative analyses in the LLM's activations.

Calling this method In-Context RL does not seem totally accurate. RL assigns credit across steps by decomposing a return into per-step rewards. Although most of RL in LLM today avoid this and is single-step, it also inconveniently does not fit the experiments on ScienceWorld which are multi-step. Clearly the LLM is doing some form of in-context learning, but to define it as RL is likely a risky long term bet. That being said, the results of the paper are interesting enough that this controversy about terminology can be overlooked.

The comparison with Relfexion/Self-Refine leads to surprising results. Given that this baseline, if I understood correctly, also gets access to the environment reward to generated a critique, it is not totally clear why they are not closer in performance. It could actually be argued that textual is richer than scalar feedback, and should lead to improved efficiency. However, results across the board indicate this isn't the case. The quality anlysis on attention heads is particularly interesting and could be further expanded. In general, brining more understanding to the pros/cons of textual feedback versus scalar feedback is an important future direction.

**Reviewer Concerns:**

Reviewers J3p5 and iBtQ had concerns about whether the paper really presents a method that can be considered as RL. This is a fair point and it is actually unlikely that what happens during in-context can be called RL. RL indeed performs credit assignment across steps during an interaction, although recent instantiations of RL do not consider an interactive setting. There are also very little evidence that anything resembling RL algorithms happens in-context. I see this as a unfortunate misnomer by the authors. However, it is not a strong enough objection given the interesting nature of the results presented in the paper. So although this issue has not been resolved, I do not see is as enough to reject the paper.

Reviewers nrTz also had concerns about the nature of the shaded area, as well as other concerns about the specific empirical setup, which was then more clearly defined by the authors. There were also concerns about novelty: although the method is not new, the current results do show uncommon conclusions with respect to established baselines such as Reflexion.

**Reviewer Scores:**

Reviewer J3p5: It is unlikely that they would change their score.
Reviewer nrTz: It is possible that they would change their score.
Reviewer iBtQ: It is unlikely that they would change their score.

---

### Decision · Program_Chairs · 2026-01-26

Accept (Poster)